# IL-17 signalling is critical for controlling subcutaneous adipose tissue dynamics and parasite burden during chronic murine *Trypanosoma brucei* infection

Matthew C. Sinton [1,2,3] ✉, Praveena R. G. Chandrasegaran [1,2], Paul Capewell[1,2], Anneli Cooper [1,2], Alex Girard[1,2], John Ogunsola [1,2], Georgia Perona-Wright [1,4], Dieudonné M Ngoyi [5,6], Nono Kuispond[5,6], Bruno Bucheton[6,7], Mamadou Camara[6,8], Shingo Kajimura [9,10], Cécile Bénézech[11], Neil A. Mabbott [12], Annette MacLeod [1,2,4,6] & Juan F. Quintana [1,2,13,14] ✉

In the skin, *Trypanosoma brucei* colonises the subcutaneous white adipose tissue, and is proposed to be competent for forward transmission. The interaction between parasites, adipose tissue, and the local immune system is likely to drive the adipose tissue wasting and weight loss observed in cattle and humans infected with *T. brucei*. However, mechanistically, events leading to subcutaneous white adipose tissue wasting are not fully understood. Here, using several complementary approaches, including mass cytometry by time of flight, bulk and single cell transcriptomics, and in vivo genetic models, we show that *T. brucei* infection drives local expansion of several IL-17A-producing cells in the murine WAT, including $T_H17$ and $V\gamma6^+$ cells. We also show that global IL-17 deficiency, or deletion of the adipocyte IL-17 receptor protect from infection-induced WAT wasting and weight loss. Unexpectedly, we find that abrogation of adipocyte IL-17 signalling results in a significant accumulation of $Dpp4^+ Pi16^+$ interstitial preadipocytes and increased extravascular parasites in the WAT, highlighting a critical role for IL-17 signalling in controlling preadipocyte fate, subcutaneous WAT dynamics, and local parasite burden. Taken together, our study highlights the central role of adipocyte IL-17 signalling in controlling WAT responses to infection, suggesting that adipocytes are critical coordinators of tissue dynamics and immune responses to *T. brucei* infection.

*Trypanosoma brucei* is an extracellular protozoan parasite that infects humans and livestock, causing Human African Trypanosomiasis (HAT, or sleeping sickness) and Animal African Trypanosomiasis (AAT, or nagana), respectively[1]. Both HAT and AAT are prevalent in sub-Saharan regions of the African continent, where they impose a significant socio-economic burden, and are fatal if left untreated[2]. Chronic infections in both humans and non-primate mammalian hosts, such as

domestic cattle, lead to significant weight loss, a phenomenon that remains largely unstudied[3]. Upon infection, trypanosomes proliferate and migrate into tissues throughout the body, where they persist and form extravascular reservoirs in virtually every organ[4]. One major consequence of infection is weight loss, typically coupled with a reduction in white adipose tissue (WAT) mass[5]. Previous studies have elegantly demonstrated that mice lose weight during *T. brucei*

infection and that this is associated with loss of gonadal white adipose tissue (gWAT) mass[6]. The largest cell volume within the gWAT is comprised of adipocytes, but this tissue is also enriched with multiple immune cell types, including macrophages, neutrophils, T helper 1 ($T_H$1) cells, effector CD8[+] cytotoxic T cells and B cells[7–9], as well as forming a reservoir for *T. brucei*, and different factors released from these immune cells influence gWAT function under normal physiological conditions. For example, tumour necrosis factor (TNF) and interleukin-17A (IL-17A) have been shown to regulate adipose tissue structure and function, by limiting tissue expansion[7], and inhibiting adipogenesis[9], respectively. Furthermore, IL-17A, and signalling through the IL-17C receptor, have been shown to induce thermogenesis in white and brown adipose tissue, respectively[10,11], and activation of thermogenesis, following challenges such as cold exposure, leads to increased energy expenditure[12]. *T. cruzi* and *T. congloense* infections, the causative agents of Chagas disease and AAT, respectively, are also associated with elevation of IL-17A[13,14], which is important for controlling resistance to infection[14,15]. Together this may suggest that loss of adipose tissue mass and subsequent weight loss may be driven by local immune responses, the parasites, or both. However, it remains unclear how the immune response to *T. brucei* infection influences adipose tissue structure and function.

Previous studies from our lab identified the skin as another reservoir for *T. brucei* and highlighted the presence of parasites in the adjacent subcutaneous white adipose tissue (scWAT) of infected patients[16]. Furthermore, we recently demonstrated a critical role for γδ T cells (in particular IL-17-producing Vγ6[+] cells) in the control of local skin responses to *T. brucei* infection[17]. However, we were unable to profile the scWAT in detail. Due to its proximity to the skin, the scWAT may also prove to be an important reservoir for *T. brucei*, giving the parasites access to a plentiful nutrient supply, whilst simultaneously increasing the chances of onward transmission. Therefore, we focused on understanding the impact of infection on the structure and function of the inguinal white adipose tissue (iWAT) in mice, which is typically used to model scWAT in humans[18]. Like gWAT, the iWAT acts as an energy reservoir under homeostatic conditions and modulates systemic metabolism and appetite[19], as well as immune responses[20].

Here we present data demonstrating that *T. brucei* infection is associated with a broad immune response in the iWAT, including the expansion of IL-17-producing $T_H$17 and Vγ6[+] cells. Furthermore, we demonstrate that global genetic ablation of IL-17A/F or targeted deletion of adipocyte IL-17A receptor (*Il17ra*; essential for mediating IL-17A/F signalling[21]) prevents or limits weight loss and iWAT wasting. Targeted deletion of adipocyte *Il17ra* also results in an accumulation of small adipocytes in both naive and infected mice, with a concomitant accumulation of *Dpp4[+] Pi16[+]* interstitial preadipocytes. Interestingly, targeted deletion of adipocyte *Il17ra* also resulted in an increased burden of extravascular parasites in the iWAT. These results provide insights into the role of IL-17 signalling as a regulator of adipose tissue structure, function, and dynamics during infection, placing adipocyte-mediated responses at the core of the local immune responses in the iWAT. Furthermore, these findings support the utility of *T. brucei* infection models for interrogating the role that IL-17 signalling plays in controlling adipocyte fate and differentiation, as well as local and systemic energy balance, in the context of infection.

## Results

### *T. brucei* infection results in reduced iWAT mass and impaired adipose tissue function in a sex-dependent manner

In both humans and livestock, trypanosome infections are known to cause weight loss, and this has been recapitulated in male mouse models of infection[6]. However, direct comparisons have not been made between male and female mice to assess whether infection induces weight loss in a sexually dimorphic manner. To test this, we infected age-matched male and female C57BL/6 mice for a period of 25 days. We first wanted to determine that mice were successfully infected and whether there were differences between the levels of circulating parasites between sexes. Parasitaemia measurements followed a characteristic pattern, with no significant differences between sexes (Fig. 1A). There were also no significant differences in the clinical scores of the mice (Fig. 1B). Strikingly, during the course of infection, infected male mice lost significant amounts of bodyweight (Fig. 1C), whereas there was no significant difference between the weights of naive and infected female mice (Fig. 1D). Spleen mass increased similarly in both male and female mice (Supplementary Fig. 1), suggesting that changes in spleen mass during infection do not explain the differences in the bodyweight of males and females.

Weight loss may be explained as a consequence of adipose tissue wasting[22], as a consequence of changes in feeding behaviour[23], or both. To understand this in more detail, we measured gross food intake as a proxy for feeding behaviour. Over the course of infection, the food intake of infected male mice decreased from the onset of infection until 11 days post-infection (dpi), after which it increased to that of naive males, before dropping again at 25 dpi (Fig. 1E). Infected female mice displayed a brief, but non-significant, reduction in food intake at 7 dpi, but otherwise maintained a similar feeding behaviour profile to their naive counterparts (Fig. 1F). Taken together, these data suggest that weight loss during experimental trypanosomiasis occurs in a sex-dependent manner and may be associated, in males, to changes in feeding behaviour.

We next explored the impact of *T. brucei* infection on the adipose tissue. In addition to changes in feeding behaviour[23], weight loss is typically associated with reductions in adipose tissue mass[24]. We were particularly interested in the iWAT, which is analogous to the scWAT tissue in humans, as this constitutes an important parasite niche for disease transmission, especially in asymptomatic carriers[16]. Previous reports have shown that colonisation of the gonadal white adipose tissue (gWAT) is associated with weight loss and reduction in adipose mass[6], but the effect on iWAT, which is anatomically and functionally distinct from gWAT (Fig. 2A), was not investigated. We first determined the presence of parasites in the iWAT and gWAT by histological analysis (Fig. 2B) and, as expected, detected trypanosomes in both tissues. Next, we quantified trypanosome genomic DNA in the iWAT and gWAT as a proxy for determining parasite density. We found that in male mice there were fewer parasites in the iWAT compared with the gWAT, but that there were no differences between these depots in females (Fig. 2C). Our data are consistent with previous reports[6], and highlights that the iWAT also harbours a population of parasites that, given their proximity to the skin, may be important for forward parasite transmission[16].

Following our observations of weight loss and the presence of trypanosomes in the iWAT, we then proceeded to characterise the impact of infection on this adipose tissue depot. When normalised to bodyweight, we found that infection led to a significant reduction in the iWAT mass of male mice (Fig. 2D). Female mice also experienced a reduction in iWAT mass, but this was not significant. This raised the question of whether the reduction in mass is due to a loss of lipid content and reduction in adipocyte size (hypotrophy). To address this, we performed Haematoxylin and Eosin (H&E) staining of iWAT at 25 dpi. We found that the iWAT of male and female mice infected with *T. brucei* undergoes hypotrophy, with concurrent infiltration of immune cells (Fig. 2E). Furthermore, we found a differential effect on the morphometric properties of the iWAT adipocytes in response to *T. brucei* infection. In infected male mice, there was a significant decrease in lipid droplet size, with the majority of lipid droplets ranging from 10 to 200 µm², compared with naive controls where lipid droplet size was larger, with the majority of droplets ranging from 51 to 900 µm² (Fig. 2F). In infected female mice, there was a less dramatic shift in lipid droplet size, with the majority of lipid droplets ranging from 10 to 400 µm², compared with naive controls where lipid droplet sizes

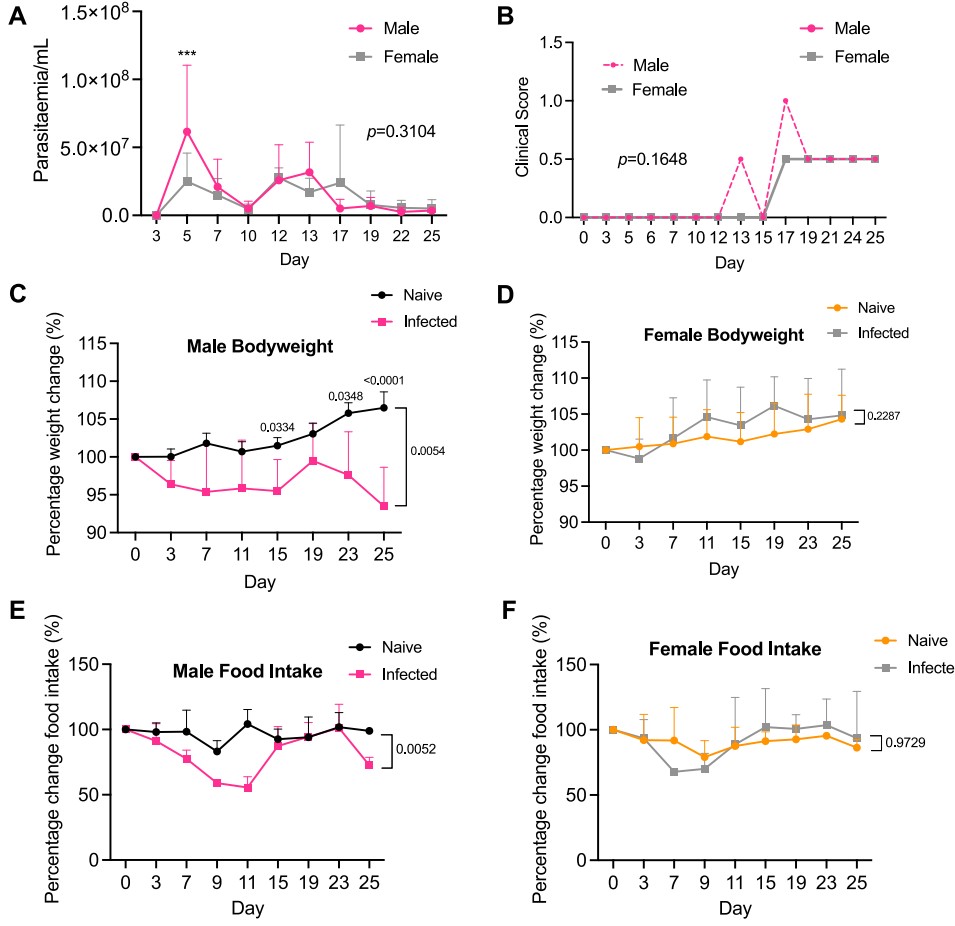

**Fig. 1 | Male mice lose weight during infection with *Trypanosoma brucei*, which is associated with alterations in feeding behaviour. A** Number of parasites per mL of blood, measured using phase microscopy and the rapid "matching" method[81]. $N = 8$ biological replicates per group from two independent experiments. **B** Clinical scores of infected male and female mice. **C** Percentage changes in bodyweight of male ($n = 5$ naive and $n = 8$ infected biological replicates per group across two independent experiments) and **D** female mice over the course of infection ($n = 8$ naive and $n = 11$ infected biological replicates per group across 2 independent experiments). Percentage changes in gross food intake in male (**E**) and female (**F**) mice. Each data point represents gross food intake from two cages ($n = 3$ naive mice per cage and $n = 4$ infected mice per cage). Time series data were analysed using two-way repeated measures ANOVA with Sidak post hoc testing and are expressed as mean ± SD. iWAT inguinal white adipose tissue, gWAT gonadal white adipose tissue.

ranged from 51 to 1000 μm² (Fig. 2G). The morphometric analyses of iWAT in response to infection are indicative of tissue wasting. Thus, we questioned whether *T. brucei* infection impacted adipose tissue function, using systemic glycerol levels as a proxy for adipose tissue function[25]. In both infected male and female mice, the circulating glycerol levels were significantly reduced compared to naive controls (Fig. 2H and Supplementary Data 1), consistent with a global impact of infection on adipose tissue function. The changes in glycerol levels observed in experimental infections were also replicated in the serum of stage II HAT patients from the towns of Boffa, Forécariah and Dubréka in Guinea, suggesting that the adipose tissue dysfunction induced by infection also occurs in humans (Fig. 2I). Taken together, these findings highlight that *T. brucei* infection is associated with significant iWAT wasting and that this, in turn, is associated with impaired tissue function in both mice and humans.

### *T. brucei* iWAT colonisation results in a transcriptional profile indicative of energy conservation in iWAT

To better understand the iWAT response to infection, and to identify potential drivers of tissue wasting, we performed bulk transcriptomic analysis of iWAT harvested at 25 dpi from both sexes and included naive controls for comparison. Principal component analysis (PCA) revealed high levels of variance between naive and infected male and

female mice (Fig. 3A). Compared with naive controls, differential expression analysis revealed upregulation of 3828 and 3177 genes with a log₂Fold change >0.5 and an adjusted *P* value (*P*adj) of <0.01 in infected male and female mice, respectively (Fig. 3B and Supplementary Data 2). In contrast, compared with naive controls, this analysis revealed downregulation of 3332 and 2606 genes an absolute log₂ fold change >0.5 and a *P*adj of <0.01 in infected male and female mice, respectively (Fig. 3B and Supplementary Data 2). To explore this dataset further, we performed pathway enrichment analyses, enriching for Kyoto Encyclopaedia of Genes and Genomes (KEGG) terms. We first observed that in both males and females, the majority of upregulated pathways were associated with immune and inflammatory processes (Fig. 3C, D). Several genes within these pathways related to antigen processing and presentation (KEGG mmu04612; e.g., *H2-DMa, H2-DMb1, H2-Ob*), cytokine-cytokine receptor interactions (KEGG mmu04060; e.g., *Tnf, Ifng, Il1b*), and complement and coagulation cascades (KEGG mmu04610; e.g., *C2, C3, C4b*) (Supplementary Data 2 and 3). When we explored downregulated pathways, we found that the majority of these were related to metabolism (Fig. 3E, F), including valine, leucine and isoleucine degradation (KEGG mmu00280; e.g., *Acat1, Acat2, Abat*) (Supplementary Data 2 and 3) and propanoate metabolism (KEGG mmu00640; e.g., *Sucla2, Acox1, Acads*) (Supplementary Data 2 and 3). Unexpectedly, we also observed that *T. brucei*

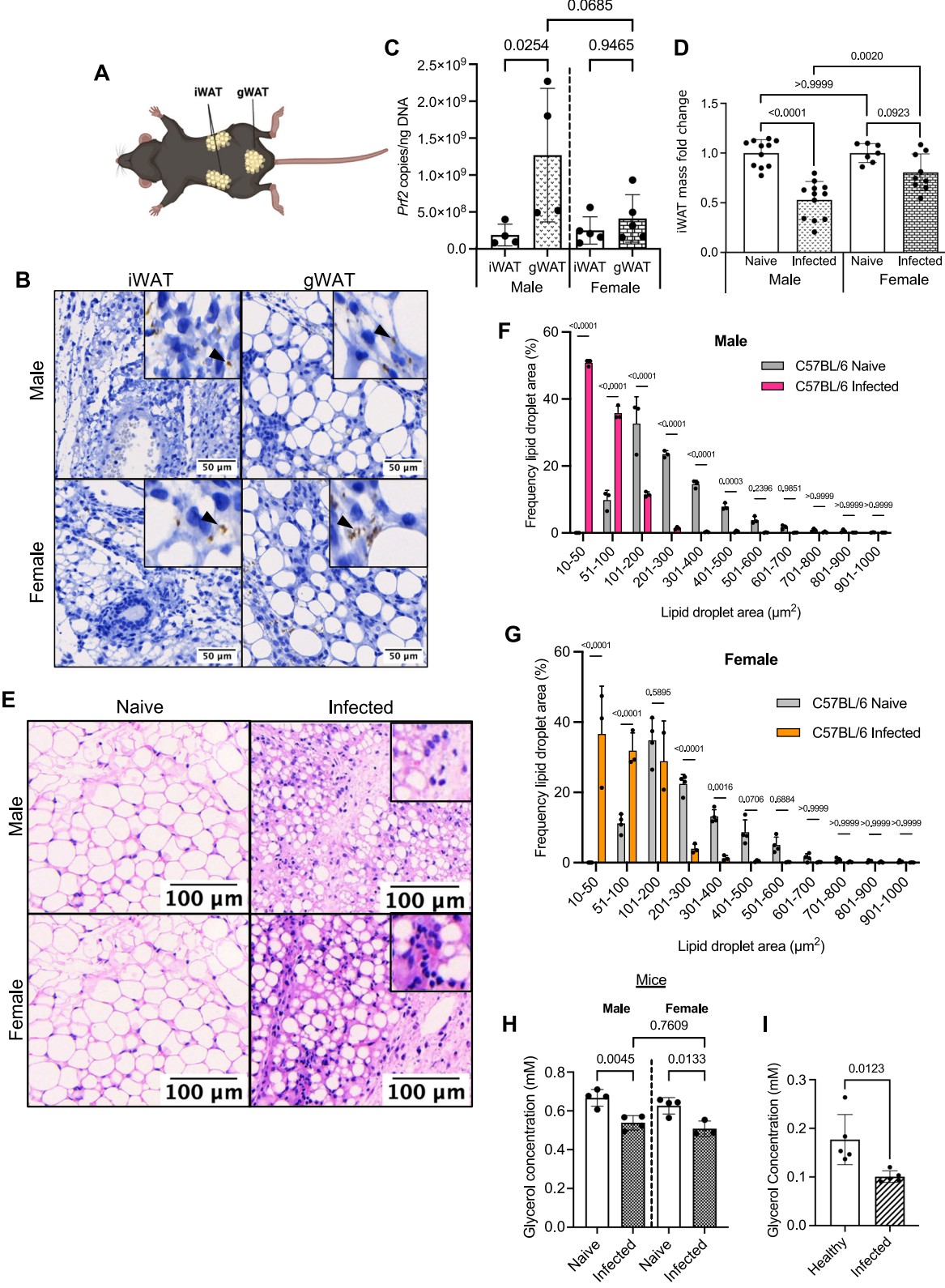

infection led to a downregulation of the lipolysis pathway in males and females (KEGG mmu04923) (Supplementary Data 2 and 3), with canonical genes such as *Pnpla2*, *Fabp4* and *Lipe* being significantly downregulated (Supplementary Fig. 2A, B). Since lipolytic genes were downregulated, we questioned whether the iWAT was becoming more thermogenic, which could, in turn, lead to a decrease in mass. However, when exploring our data, we also found that canonical genes associated with thermogenesis, such as *Ucp1*[26], were downregulated in

infected males and females (Supplementary Data 2 and 3), Moreover, expression of genes associated with UCP1-independent thermogenesis, including *Atp2a1*, *Atp2a2*, and *Atp2a3*[27] were not altered during infection (Supplementary Data 2 and 3), suggesting that thermogenesis is not activated in this context. The downregulation of genes associated with energy metabolism pathways, such as lipolysis and thermogenesis, together with an overall decrease in circulating glycerol levels in serum, could suggest that chronic *T. brucei* infection

**Fig. 2 | *Trypanosoma brucei* infection leads to reductions in iWAT mass and lipid content, as well as impairment of adipose tissue function. A** Schematic highlighting the anatomical location of the inguinal white adipose tissue (iWAT) and gonadal white adipose tissue (gWAT). Created with Biorender (Agreement number ZF25XVXG0V). **B** Histological analysis of the iWAT and gWAT trypanosome colonisation, using HSP70 staining. **C** Parasite burden of iWAT and gWAT. Parasite density in the iWAT, which was measured by RT-qPCR of genomic DNA. A comparison was made with gonadal white adipose tissue (gWAT), to understand if the iWAT is also highly colonised. $N = 4$ biological replicates collected from one experiment. **D** iWAT mass at 25 days post-infection or in naive mice. iWAT was dissected and weighed before normalising to bodyweight, to account for variation between biological replicates. Symbols indicate the number of biological replicates collected from two independent experiments. $N = 11$ (naive male), 12 (infected male), 7 (naive female) and 9 (infected female) biological replicates from two independent experiments. **E** Representative histological H&E staining of iWAT showing adipocyte lipid droplets and immune infiltrate. Insets highlight likely immune cell infiltrate. **F** Analysis of lipid droplet area (μm²) in naive and infected males and **G** females. $N = 3$ biological replicates per group. Lipid droplets were measured from three distinct areas in each image and then combined for each biological replicate. **H** Circulating glycerol concentration in naive vs. infected male and female mice. $n = 3–4$ biological replicates per group. **I** Circulating glycerol concentration in healthy vs. infected patients. $N = 5$ patients per group and each group is a mix of male and female. **C, D, H, I** Data were analysed using a two-tailed Student $t$ test. **F, G** Data were analysed using a two-way ANOVA with Sidak post hoc testing. Data for all panels are expressed as mean ± SD.

leads to mice (and likely humans) entering a state of energy conservation[28].

### *T. brucei* iWAT colonisation induces upregulation of genes associated with T$_H$17 T cell responses

Our transcriptomic analyses also revealed specific enrichment of T cell-related transcripts in male but not female mice. For example, we found that *Cd3d*, *Cd3e*, and *Cd247* were significantly upregulated in the iWAT of infected male mice but exhibited no changes in infected females (Supplementary Data 3). T$_H$1- and T$_H$2-related transcripts included *Nfatc1*, *Cd4*, *Runx3*, and *Gata3*, suggesting that T$_H$1 cells may be a significant contributor to interferon-gamma (IFNγ) production during infection in the adipose tissue. Additionally, we also detected significant upregulation of genes associated with the differentiation of T$_H$17 effector cells, including *Irf4*, *Cd4*, *Il21r*, and *Il6ra* in infected male mice but not in females (Fig. 3G and Supplementary Data 4). Based on our observations, we hypothesised that T$_H$17 cells are important for the adipose immune response to infection. To quantify the different populations of CD4$^+$ T cells, including T$_H$17 cells, present in the iWAT during chronic *T. brucei* infection, we utilised mass cytometry by time of flight (CyTOF), enabling us to gain as much information as possible from the wasted adipose tissue. Consistent with previous studies[8], we also found an expansion of macrophages, granulocytes, and dendritic cells in the iWAT of infected males and female mice (Fig. 4A–D). The B cell subset was decreased specifically in male mice during infection compared to naive controls, without noticeable changes in female mice (Fig. 4E). Regarding the T cell effector population, we observed an increase in the proportion of CD3ε$^+$ TCRβ$^+$ CD4$^+$ T cells in the iWAT of infected mice (Fig. 4F) and, in particular, we identified an expansion of CD44$^+$ CD69$^+$ CD4$^+$ T effector (Teff) cells (Fig. 4G). Whilst the frequency of CD4$^+$ T cells increased in both males and females, the iWAT of infected females contained a higher proportion of Teff cells compared with males. The expanded Teff cells displayed an elevated production of interferon-gamma (IFNγ; Fig. 4H), suggesting that some of these cells are polarised towards a T$_H$1 phenotype. Furthermore, there was a significant expansion of IL-17A-producing Teff cells (Fig. 4I), consistent with the transcriptional responses of the iWAT to infection. Interestingly, in addition to TNFα and IFNγ, IL-17A was also elevated in the serum of infected mice and compared to naive controls (Fig. 4J), and in first and second-stage HAT patients compared to healthy controls (Fig. 4K and Supplementary Data 1), suggesting that IL-17A elevation is conserved in mice and humans infected with *T. brucei*. Although we observed an increase in the frequency of IL-17A$^+$ Teff cells and circulating IL-17A in both males and females, we only observed an upregulation of the cognate IL-17A receptor (*Il17ra*) in the iWAT of male mice during infection (Fig. 4L). This could suggest that the iWAT becomes more responsive to IL-17A signalling in males than females during infection. Taken together, our data demonstrate a significant expansion of IL-17-producing T cells in the iWAT in response to *T. brucei* infection. Moreover, our data could suggest that the iWAT of

male mice becomes more responsive to IL-17A signalling during *T. brucei* infection, compared with females.

### IL-17A/F is critical for controlling bodyweight and pathology during *T. brucei* infection

Our results so far indicate that chronic iWAT infection leads to an expansion of local IL-17A-producing Teff cells. However, the role of IL-17 in the control of *T. brucei* infection, or in the local infection-induced pathology in the iWAT, has not been explored to date. To test this, we used a global *Il17a/f* knockout mouse, which is deficient in both IL-17A and IL-17F[29] (Fig. 5A and Supplementary Fig. 3A). Systemically, we found that there were no significant differences in parasitaemia when comparing C57BL/6 and *Il17af$^{-/-}$* male (Fig. 5A). In contrast, the first peak of parasitaemia in *Il17af$^{-/-}$* females was lower than in C57BL/6 mice, and the second peak of parasitaemia was delayed (Supplementary Fig. 3A). Deficiency of *Il17a/f* was also associated with an earlier onset and increased severity of clinical symptoms in both males (Fig. 5B) and females (Supplementary Fig. 3B). *Il17af$^{-/-}$* male and female mice started to exhibit clinical symptoms (piloerection and hunching) from 3 and 7 dpi, respectively, whereas C57BL/6 mice started to experience these symptoms between 12 and 15 dpi. The parasite burden of the major adipose tissue depots does not appear to be influenced by IL-17A/F in either sex (Fig. 5C and Supplementary Fig. 3C, D), indicating that IL-17A/F is dispensable for controlling local parasite burden in the adipose tissue. We only detected significant differences in the dynamics of circulating parasites in *Il17af$^{-/-}$* female mice, which displayed a reduced initial peak of parasitaemia at 5dpi, and a delayed second peak of parasitaemia at 15 dpi compared to C57BL/6 controls (Supplementary Fig. 3A).

Having established that IL-17A/F signalling is critical for controlling clinical symptoms during *T. brucei* infection, we next examined the role of IL-17A/F in controlling bodyweight and local iWAT pathology. For this, we monitored the weight of both male and female mice during the course of infection. Unlike C57BL/6 male mice, infected *Il17af$^{-/-}$* males increased their bodyweight (-8–10%) during infection (Fig. 5D). Infected female *Il17af$^{-/-}$* mice also gained weight during infection (-8–19%) in a similar pattern to C57BL/6 females, although similarly to males, IL-17A/F-deficient females gained more weight than their C57BL/6 counterparts (Supplementary Fig. 3E). Previous reports have shown that when administered to C57BL/6 naive mice under homeostatic conditions, IL-17A suppresses food intake[30]. Therefore, given the potential link between IL-17 signalling, bodyweight, and food intake, we next measured gross food intake in infected C57BL/6 and *Il17af$^{-/-}$* mice. We observed that up until 9 dpi, C57BL/6 and *Il17af$^{-/-}$* male mice reduced their food intake at the same rate (Fig. 5E). However, after 11dpi, *Il17a$^{-/-}$* mice started to increase their food intake above that of the C57BL/6 mice. Indeed, food intake for infected *Il17af$^{-/-}$* mice was higher between 15 and 23 dpi compared with at the onset of infection. Unlike male mice, food intake was indistinguishable between infected female *Il17af$^{-/-}$* and C57BL/6 mice (Supplementary Fig. 3SF). Together, our results suggest that IL-17A/F potentially regulates bodyweight by

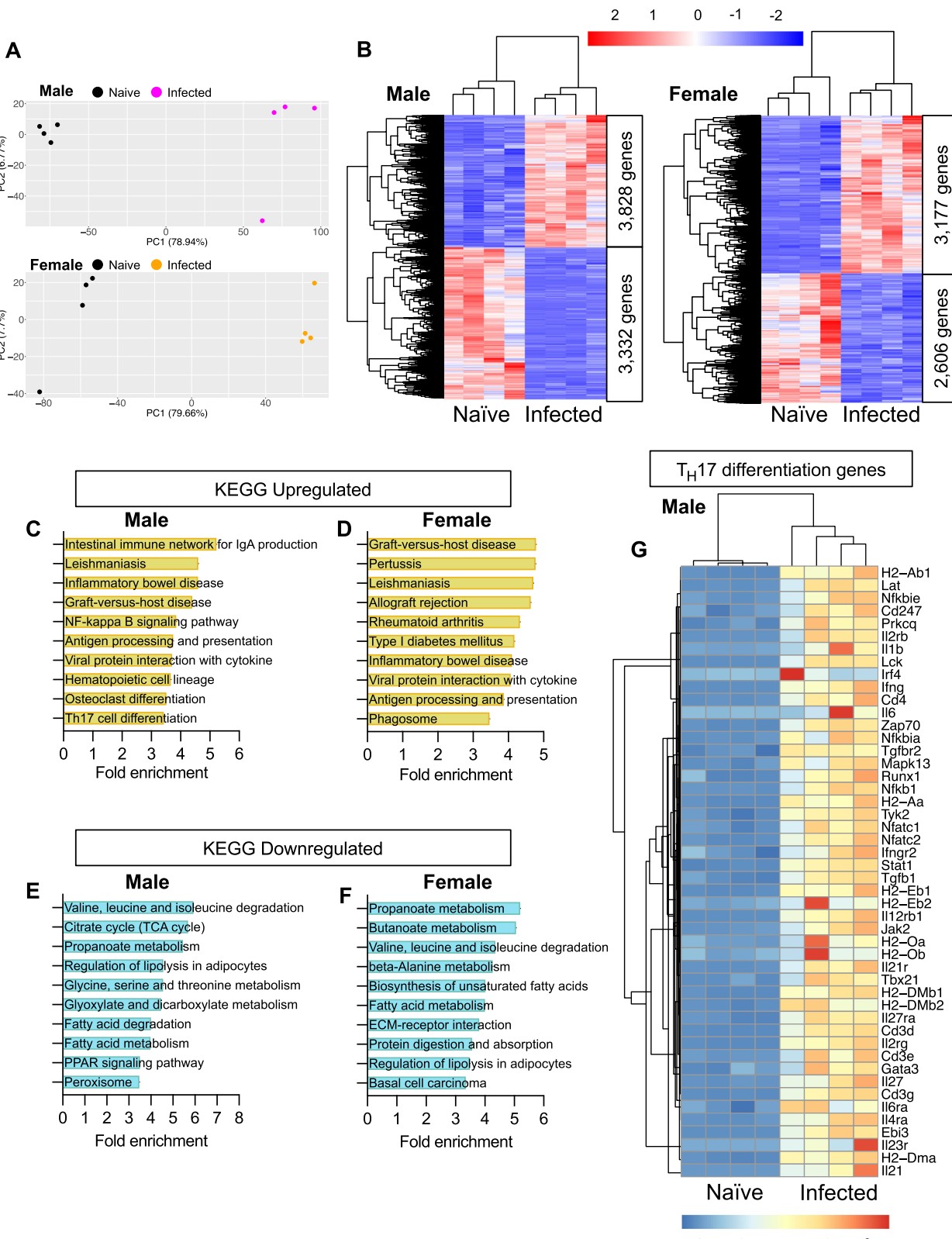

**Fig. 3 | Transcriptomic analyses indicate a stronger T$_H$17 response in the iWAT of males than females. A** Principal component analysis of bulk transcriptomic data from male and female naive and infected mice. **B** Heatmaps, clustered by Euclidean distance, of differentially expressed genes in the iWAT of naive vs infected male and female mice, scaled by Z-score. **C, D** Pathway enrichment analysis of upregulated and downregulated genes in male mice. **E, F** Pathway enrichment analysis of upregulated and downregulated genes in female mice. **G** Heatmap of T$_H$17 transcripts in male mice. For all panels, $n = 4$ biological replicates per group from a single experiment.

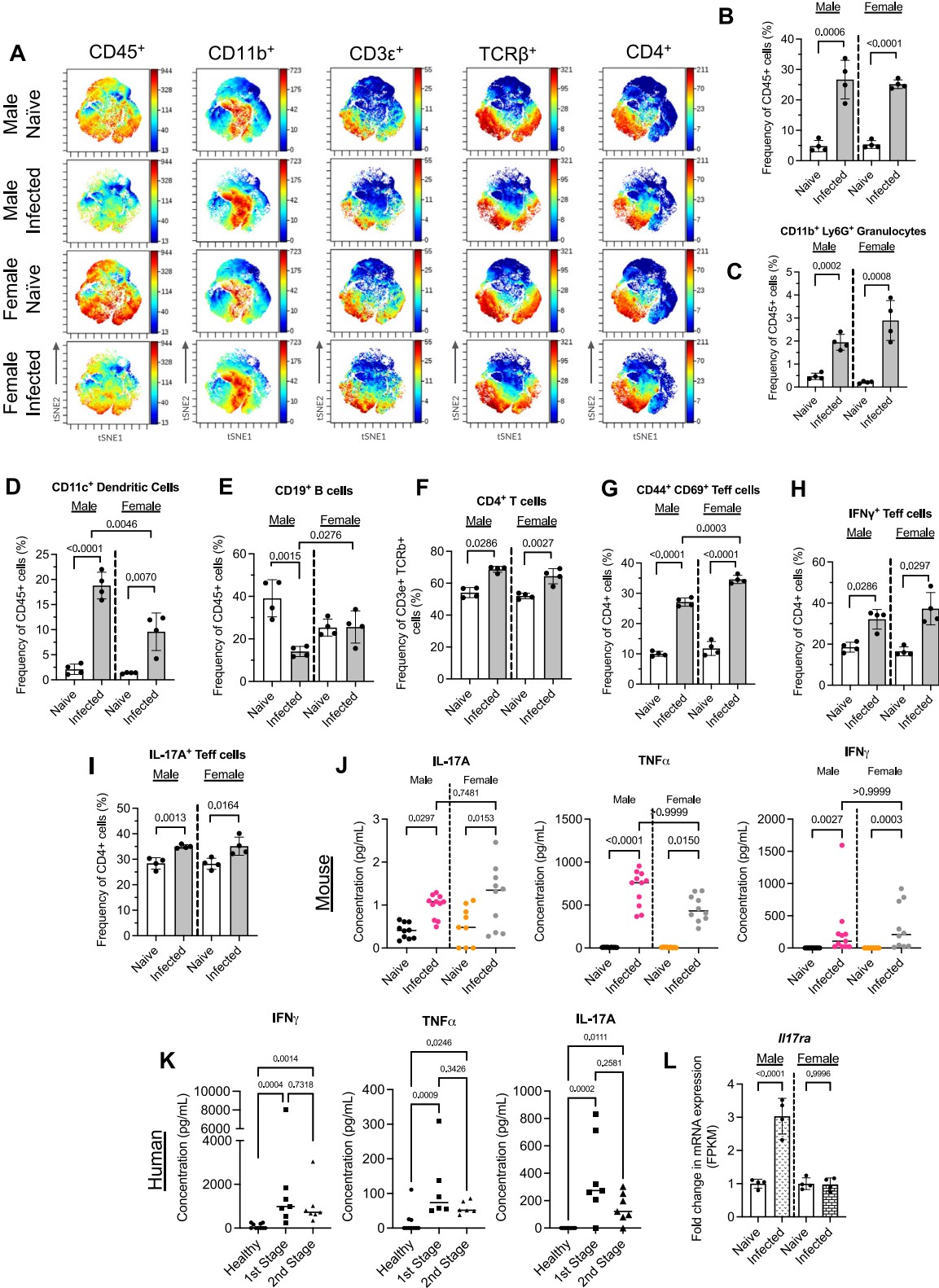

**Fig. 4 | Elevated circulating IL-17A is associated with increased IL-17 receptor A (IL-17RA) in male mice during *T. brucei* infection. A** t-distributed stochastic neighbour embedding (t-SNE) plots of broad macrophage and T cell populations. We gated on CD45+, CD3ε+, TCRβ+, CD4+ cells. **B** Measurements of the proportions of macrophages, **C** granulocytes, **D** dendritic cells, and **E** B cells. **F** Measurements of the proportions of CD4+, **G** T effector (Teff), **H** IFNγ-producing cells, **I** IL-17A-producing Teff cells. Measurements of circulating serum IL-17A, TNFa, and IFNγ in mice (**J**) and IFNγ, TNFa and total IL-17 in humans (**K**). **L** Expression of interleukin-17A receptor (*Il17ra*) mRNA in infected male and female mice. Mouse cytokine data were collected from samples taken across three independent experiments. Data were tested for normal distribution and analysed by either one-way ANOVA or a Kruskal Wallis test. Biological replicates are indicated by symbols for each panel. Data for all panels are expressed as mean ± SD.

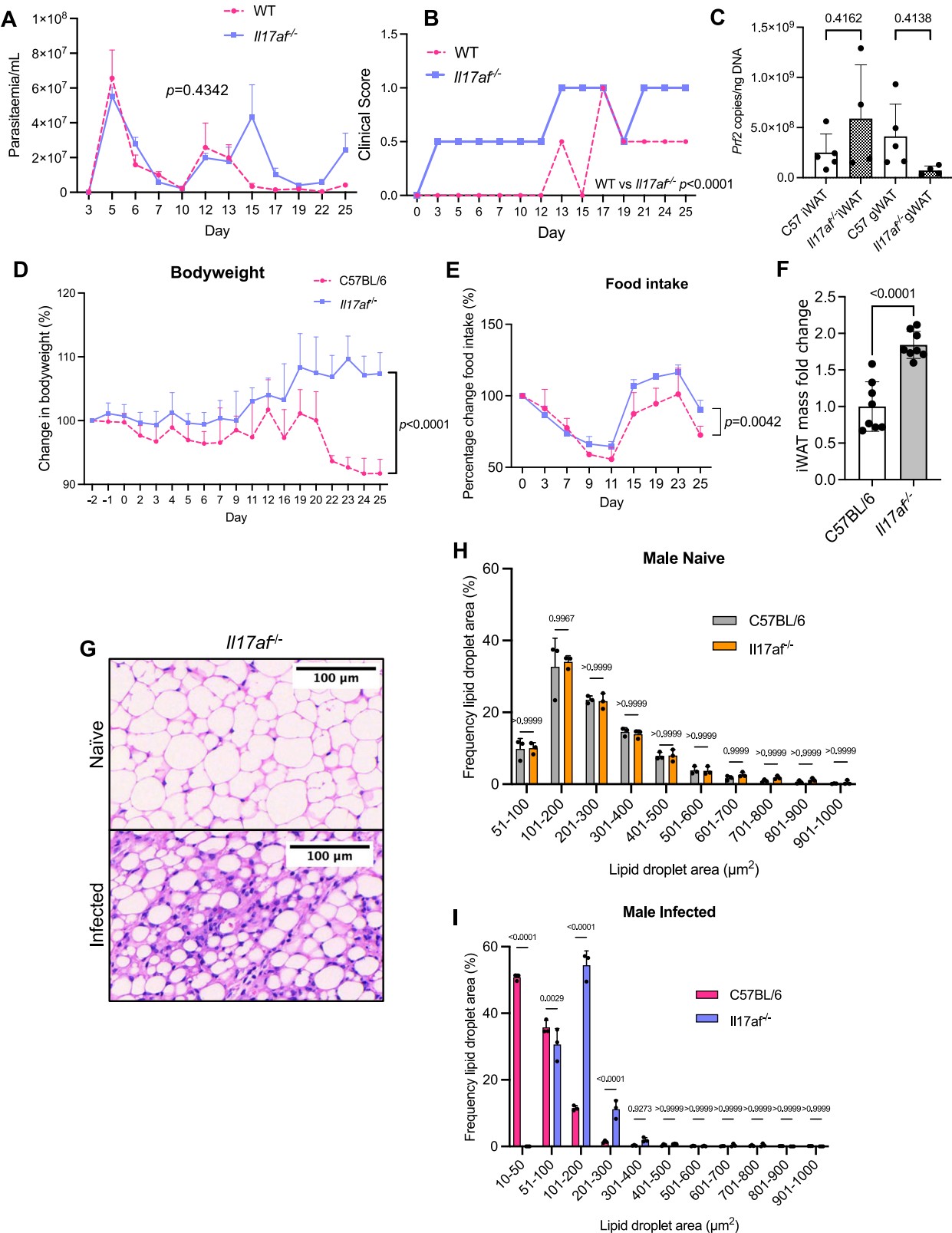

altering food intake in male mice during infection, but not female mice. We observed that the weight gain in females was also associated with increased splenomegaly in *Il17af⁻/⁻* mice, compared with their C57BL/6 counterparts or to male mice, thus indicating that splenomegaly by itself does not account for the increase in bodyweight observed in male mice (Supplementary Fig. 3G, H). Related to the differences in bodyweight, we also found that upon infection, *Il17af⁻/⁻* knockout male

mice retained more of their iWAT mass compared with their 57BL/6 counterparts (Fig. 5F), suggesting that they experienced less wasting. In contrast, although naive *Il17af⁻/⁻* female mice have a higher bodyweight than their C57BL/6 counterparts, they experience similar levels of wasting as C57BL/6 females upon infection (Supplementary Fig. 3I), supporting our hypothesis that the effects of IL-17A/F are sex-dependent.

**Fig. 5 | IL-17A/F limits clinical symptoms during *T. brucei* infection and drives weight adipose tissue wasting in male mice. A** Number of parasites per mL of blood in *Il17af⁻/⁻* vs C57BL/6 male mice, which was measured using phase microscopy and the rapid "matching" method[81]. *N* = 8 biological replicates across two independent experiments. **B** Comparison of clinical scores between C57BL/6 and *Il17af⁻/⁻* mice. **C** Parasite burden of inguinal WAT (iWAT) and gonadal WAT (gWAT). *N* = 5 (C57BL/6 iWAT and C57BL/6 gWAT and *n* = 4 (*Il17af⁻/⁻* iWAT and *Il17af⁻/⁻* gWAT) biological replicates per group from one experiment. **D** Percentage changes in bodyweight of C57BL/6 and *Il17af⁻/⁻* male mice over the course of infection. *N* = 7 mice per group across two independent experiments. **(E)** Changes in gross food intake over the course of infection. Each data point represents gross food intake

from two cages (*n* = 3 naive mice per cage and *n* = 4 infected mice per cage). **F** iWAT mass in infected C57BL/6 and *Il17af⁻/⁻* male mice. *N* = 8 biological replicates per group across two independent experiments. **G** H&E staining of iWAT from male *Il17af⁻/⁻* mice. **H, I** Analysis of lipid droplet area (μm²) in naive and infected male mice. Lipid droplets were measured from three distinct areas in each image and then combined for each biological replicate. *N* = 3 biological replicates per group from one experiment. Data for C57BL/6 mice in (**A, B, D**) are taken from Fig. 1. Data for C57BL/6 mice in (**H**) are taken from Fig. 2. Data were analysed using two-way repeated measures ANOVA with Sidak post hoc testing or a one-way ANOVA with Tukey post hoc testing. Data points represent biological replicates. Data for all panels are expressed as mean ± SD. iWAT inguinal white adipose tissue.

To understand whether global deficiency of *Il17af⁻/⁻* impacts iWAT lipid content, we performed H&E staining on the iWAT and measured lipid droplet size in naive and infected male (Fig. 5G–I) and female (Supplementary Fig. 3J–L) mice and compared these with C57BL/6 animals. The iWAT adipocyte size was indistinguishable between genotypes in naive animals, indicating that IL-17A/F does not affect lipid droplet size under homeostasis. However, upon infection, *Il17af⁻/⁻* males retained a higher frequency of larger lipid droplets than males (Fig. 5I). When comparing this to female mice, we found that deletion of *Il17af⁻/⁻* had no impact on the range of lipid droplet sizes during *T. brucei* infection (Supplementary Fig. 3L). Our results indicate that IL-17A/F signalling drives iWAT wasting and lipid usage in adipocytes in males, but not in females, during *T. brucei* infection.

### Single-cell analysis of the iWAT stromal vascular fraction reveals a distinct population of IL-17-producing CD27⁻ Vγ6⁺ cells in the iWAT of infected mice

To further understand the events leading to iWAT wasting in response to *T. brucei* infection, we conducted single-cell RNA on the iWAT, now focusing on male mice since this is where we observed both upregulation of the IL-17A receptor (Fig. 4L) and changes in bodyweight (Fig. 5D). As we wanted to understand the early processes leading up to the loss of iWAT tissue mass, we infected mice for 7 days before harvesting the iWAT. Following iWAT dissociation and scRNAseq analysis, we obtained a total of 46,546 high-quality cells with an average of 1296 genes and 20,209 reads per cell (Supplementary Data 5). These cells were clustered at a resolution of 0.6 using Clustree analysis (Supplementary Fig. 4A). Cells were then broadly classified into 18 clusters (Fig. 6A and Supplementary Data 6) based on common markers associated with these clusters. The predominant cell type in this dataset was immune cells, with four B cell clusters, two plasma cell clusters, four T cell clusters, an NKT cell cluster, two myeloid clusters, and a plasmacytoid dendritic cell (pDC) cluster (Fig. 6A). In addition to the cells within the immune compartment, we identified two stromal cell clusters comprised of preadipocytes. Of the cell types we identified, we observed that several expanded in the iWAT of infected mice, including Tregs, T cell 1, B cell 4, NKT cells, and Myeloid 2, whereas other cell populations remained similar in numbers between conditions, or contracted in the case of preadipocytes, T cell 2, several B cell clusters, and pDCs (Fig. 6B). Of the B cells, we classified four clusters as B cells (B cells 1–4), based on high expression of *Cd79a Cd79b*, and *Ighd*[81], and we broadly classified plasma cells based on these markers with the addition of *Jchain* and *Ighm*[32] (Fig. 6C). We also identified a cluster of germinal centre (GC)-like B cells, which were classified based on expression of *Aicda* and *Pcna*[33]. T cells were classified as CD8a⁺ T cells (*Cd3d⁺, Cd3e⁺, Cd8a⁺, Trac⁺*)[34], Tregs (*Cd3d⁺, Cd3e⁺, Cd4⁺, Foxp3⁺*), NKT cells (*Cd3d⁺, Cd3e⁺, Gzma⁺, Gzmb⁺, Nkg7⁺, Trbc2⁺, Klrb1c⁺, Klrd1⁺*)[35], or T cell 1 and T cell 2 (*Cd3d⁺, Cd3e⁺, Trac⁺*)[34]. The myeloid compartment was composed of two subclusters, classified as Myeloid 1 (*Lyz2⁺, Fcer1g⁺, S100a4⁺*)[36] and Myeloid 2 (*Ccl5⁺, S100a4⁺, Tmem176a⁺, Tmem176b⁺*)[36], and we classified pDCs based on high expression of *Fcer1g, Lgals1, Siglech* and *Runx2* (Fig. 6C).

To gain further insights into the identity of the B cells that we identified in our dataset, we subclustered and reanalysed all of the B cells clusters at a resolution of 0.5 (Supplementary Fig. 5A), revealing 9 clusters. Of these clusters, we identified 6 clusters of Mature B cells (Mature B cell 1–6), based on the expression of *Cd79a, Cd79b, Cd19, H2-Aa, Sell, Ltb, Ighd*, and *Ighm*[37,38] (Supplementary Fig. 5B). We also identified replicative germinal centre (GC)-like B cells, which we classified based on the expression of high expression of *Mki67, Top2a, Pclaf, Aicda*, and *Pcna*[37,38] (Supplementary Fig. 5B). Finally, we identified two populations of plasma cells: *Ighm⁺* and *Ighg1⁺* plasma cells, which we classified based on low expression of *Cd19* and high expression of *Sdc1* and *Jchain*[37,38] (Supplementary Fig. 5B). To determine the identity of the myeloid cells in our dataset, we subset and reanalysed the Myeloid 1 and 2 and pDC clusters separately (Supplementary Fig. 6A) at a resolution of 0.3, revealing 7 clusters. We classified these as *Ly6i⁺* macrophages (*Cd68⁺, Ms4a6d⁺, Ly6i⁺, Pla2g7⁺*)[36], plasmacytoid DCs (pDCs; *Siglech⁺, Cox6a2⁺, Ccr9⁺, Cd300c⁺*)[39,40], mature DCs (*Fscn1⁺, Cacnb3⁺, Socs2⁺, Tbc1d4⁺*)[41], conventional DC2s (cDC2; *S100a4⁺, Gpr183⁺, H2-Eb1⁺, Marveld1⁺*)[42], *Mrc1⁺* macrophages (*Mrc1⁺, Folr2⁺, Pf4⁺, Fcrls⁺*)[36], neutrophils (*Ly6g⁺, Cd177⁺, Cxcr2⁺, Hdc⁺, S100a9⁺, Mmp9⁺*)[43–46], and cDC1 (*Cd8a⁺, Xcr1⁺, Gcsam⁺, Clec9a⁺, Tlr3⁺*)[42] (Supplementary Fig. 6B).

Given the prominent role of genes associated with T_H17 differentiation detected in the iWAT from infected mice detected by bulk transcriptomics (Fig. 3), we wanted to determine which cell populations were expressing *Il17a* in our dataset and identified that the Treg and T cell 2 clusters were the only clusters to express this gene (Fig. 6D). We next proceeded to resolve the intrinsic heterogeneity of the T cell subset associated with the iWAT, as we wanted to understand precisely which T cells were expressing *Il17a* and whether these cells expanded during infection, as observed at later timepoints (Fig. 4). As we were unable to resolve the precise identity of the T cell 1 and T cell 2 clusters, we subset (resolution 0.3) and reanalysed the whole T cell compartment separately (Fig. 6E, F). This resulted in five T cell subclusters, which consisted mainly of Tregs (*Cd4⁺, Icos⁺, Foxp3⁺*)[34], CD8⁺ T cells (*Cd8a⁺, Cd8b1⁺*)[34], activated NKT cells (*Nkg7⁺, Klr1d1⁺, Gzma⁺, Gzmb⁺, Nr4a1⁺*)[34], a cluster of activated and replicative T cells with evidence of TCR engagement (*Top2a⁺, Mki67⁺, Hist1h1b⁺, Nr4a1⁺*)[34], and IL-17A⁺ Vγ6⁺ cells (*Tcrg-C1⁺, Rorc⁺, Cd163l1⁺, Il17a⁺*)[47] (Fig. 6F). We were unable to resolve the identity of the replicative T cell cluster, so we reclustered and reanalysed this cluster separately (resolution 0.9), revealing 7 clusters (Supplementary Fig. 7A, B). We classified these as *Cd8a⁺* T cells (*Cd3e⁺, Cd3g⁺, Cd8a⁺, Cd8b1⁺*)[34], *Cd4⁺* T cells 1 (*Cd3e⁺, Cd3g⁺, Cd4⁺*), NKT cells (*Nkg7⁺, Klr1d1⁺, Trbc1⁺*), T_H17 cells (*Cd3e⁺, Cd3g⁺, Cd4⁺, Il17a⁺*), *Cd4⁺* T cells 2 (*Cd3e⁺, Cd3g⁺, Cd4⁺*), Tregs (*Cd3e⁺, Cd3g⁺, Cd4⁺, Foxp3⁺*), and *Il17a*-expressing Vγ6⁺ cells (*Tcrg-C1⁺, Rorc⁺, Cd163l1⁺, Il17a⁺*), consistent with our previous work[17]. T_H17 cells were classified based on upregulation of *Il17a*, which we identified using differential expression analysis comparing cells from naive and infected samples (Supplementary Data 7). Furthermore, we identified that Vγ6⁺ cells also upregulated *Il17a* during infection (Supplementary Data 7). We next examined whether there was a cell–cell communication axis between T_H17 cells and other replicative T cells found in the adipose tissue

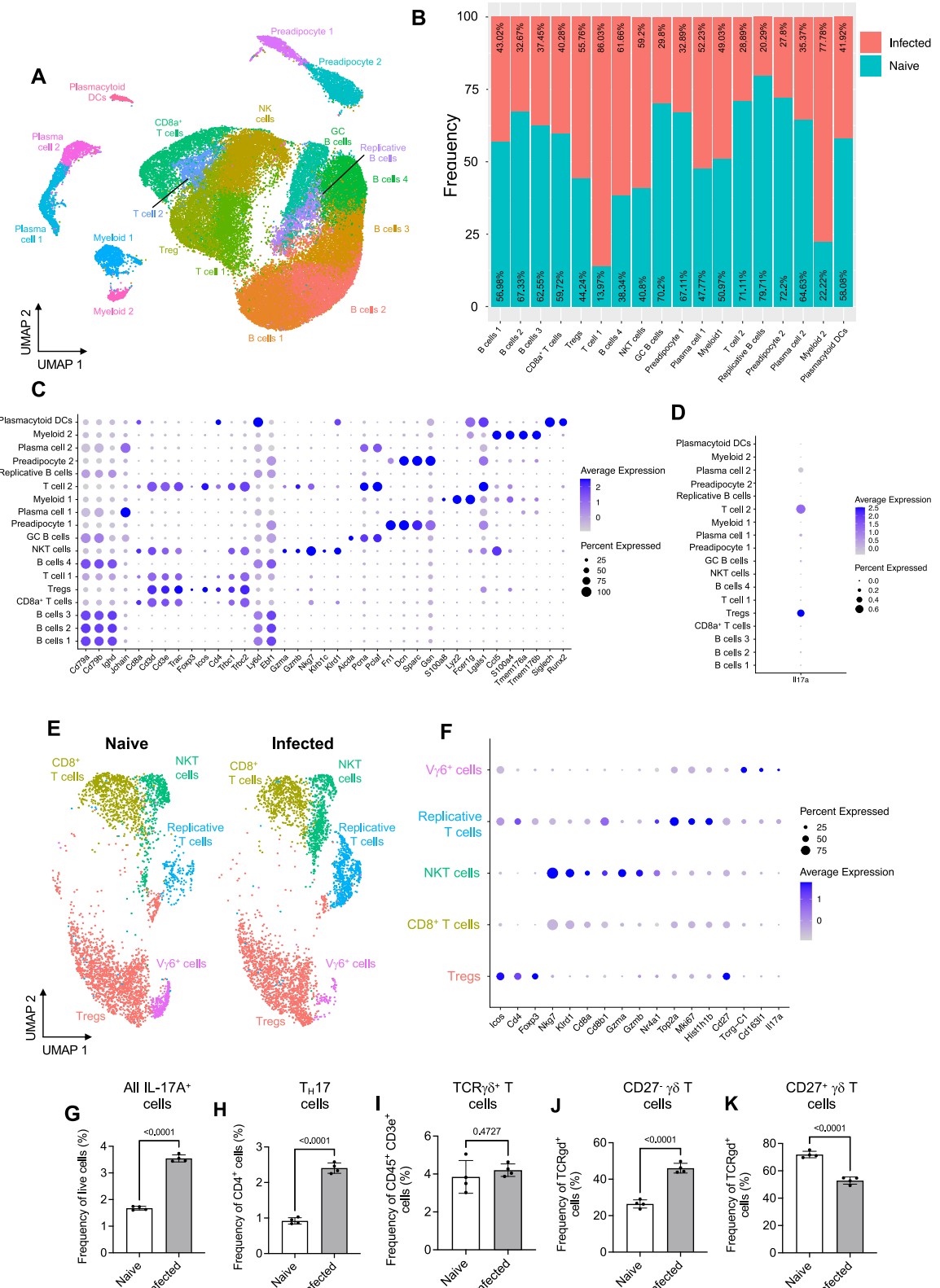

**Fig. 6 | *T. brucei* infection leads to expansion of IL-17A⁺ cells in the iWAT during early infection. A** Uniform manifold approximation and projection (UMAP) of 46,546 high-quality cells from a single-cell RNA sequencing (scRNAseq) dataset. **B** Frequency plot showing changes in cell frequency under naive or infectious conditions. **C** Dot plot representing the expression levels of top marker genes used to catalogue the diversity of cells within the dataset. The side of the dots represents the percentage of cells that express a given marker, and the colour intensity represents the level of expression. **D** Dot plot representing *Il17a* expression in cell clusters. **E** UMAP of subset T cell clusters. **F** Dot plot representing the expression

levels of top marker genes used to categorise T cell populations. **G** Flow cytometry analysis of the proportion of all live IL-17A⁺ cells. Frequency of **H** T$_H$17, and **I** TCRγδ⁺ T cells in the iWAT. Frequency of (**J**) CD27⁻ and (**K**) CD27⁺ γδ T cells in the iWAT. Single-cell data are comprised of one technical replicate per condition, each containing pooled cells from the iWAT of five male mice replicate. For flow cytometry data, data, biological replicates are indicated by symbols for each panel, and data were analysed using a two-tailed Student's *t* test. Data for all panels are expressed as mean ± SD.

during infection (Supplementary Fig. 7C). Nichenet analysis predicted that T$_H$17 cells produce several ligands that can be sensed by Tregs, including *Il17a* and *Gas6* (Supplementary Fig. 7C). Interestingly, *Gas6* interactions with *Axl* on Tregs has previously been shown to enhance the suppressive effects of Tregs, which could explain why *Il17af*$^{-/-}$ mice develop more severe clinical symptoms than C57BL/6 mice (Fig. 5B).

To validate the in silico prediction that there is an expansion of IL-17A$^+$ cells in the murine iWAT in response to infection, we next measured IL-17A expression using a murine reporter line, in which IL-17A expression is coupled to GFP expression (IL-17A$^{GFP}$; Supplementary Fig. 8). We observed that at 7 days post-infection, there was a significant expansion of IL-17A$^+$ cells (Fig. 6G and Supplementary Fig. 7), including T$_H$17 (CD45$^+$, CD3ε$^+$, CD4$^+$, GFP$^+$) cells (Fig. 6H). Although we did not observe a change in the frequency of γδ T cells (Fig. 6I), consistent with our in silico predictions, we observed a significant increase in the frequency of IL-17A-producing CD27$^-$ γδ T cells (Fig. 6J) and a concomitant decrease in the frequency of IFNγ-producing CD27$^+$ γδ T cells (Fig. 6K). Together, these findings indicate a dominant IL-17-driven response in the murine iWAT in response to *T. brucei* infection. Our data also demonstrate that IL-17A is derived from multiple sources including local T$_H$17 cells and CD27$^-$ Vγ6$^+$ cells.

### *T. brucei* infection results in broad transcriptional changes in iWAT preadipocytes, including upregulation of the IL-17A receptor and lipolysis

In addition to detecting a range of IL-17A-producing cells by scRNAseq, we also wanted to understand which cells may be responding to this cytokine during *T. brucei* infection. To this end, we measured IL-17A receptor (*Il17ra*) expression across all clusters and found that it was exclusively upregulated in preadipocytes during infection (Fig. 7A). This may indicate that of all the cells within the iWAT stroma, preadipocytes increase their responsiveness to IL-17A during infection. However, we were unable to resolve the preadipocyte heterogeneity within the iWAT, or their responses to infection, so we reclustered these cells (resolution 0.3) and reanalysed them. Using previous reports to annotate the preadipocytes[48,49], and after reclustering the population of iWAT preadipocytes (Fig. 6A), we identified five distinct populations (Fig. 7B), four of which expressed canonical mesenchymal markers including *Ly6a*, *Pdgfra* and *Cd34* (Fig. 7C)[49]. These mesenchymal subclusters encompassed interstitial preadipocytes (*Dpp4*$^+$, *Pi16*$^+$, *Bmp7*$^+$), located in the interstitium, which are known to be poised to migrate into the iWAT to differentiate into mature adipocytes when needed[49], a committed preadipocyte cluster (*Col4a1*$^+$, *Col4a2*$^+$, *Col15a1*$^+$, *Fabp4*$^+$, *Plin2*$^+$), which represent a transitional state between stem-like preadipocytes and mature adipocytes, and an adipogenesis-regulatory cell cluster (*Fmo2*$^+$, *F3*$^+$, *Clec11a*$^+$), which are able to suppress adipogenesis in vivo[50]. Lastly. We also found a cluster of mature adipocytes (*Cd36*$^+$, *Fabp4*$^+$, *Pparg*$^+$) (Fig. 7B, C). We observed that the mature adipocyte populations were exclusively detected in the naive samples and not in infected controls, perhaps due to the wasting associated with infection. During infection, we noted an increase in the frequency of the interstitial preadipocyte 1 cluster compared to naive controls (Fig. 7D), resulting in a decrease in the frequency of other subsets, including mature adipocytes, which might reflect the histological findings associated with iWAT wasting. Of the preadipocyte populations that we identified, we noted that *Il17ra* expression was upregulated in both interstitial preadipocyte clusters, and was also expressed in the committed preadipocyte cluster, (Fig. 7E and Supplementary Data 7), but not in adipogenesis-regulatory cells or mature adipocytes, suggesting that both interstitial and committed preadipocytes are able to sense IL-17A/F signalling in response to infection. Using in silico gene module scoring to assess expression of genes involved in lipolysis (e.g., *Pnpla2*, *Plaat3*, *Mgll*, *Fabp4*), we identified that responses to chronic *T. brucei* infection were observed in committed preadipocytes and adipogenesis-regulatory cells (Fig. 7F). This

possible elevation of lipolysis is in contrast to our data from later timepoints of infection, where we found evidence of decreased lipolysis (Fig. 2H, I and Supplementary Fig. 2A, B). Glycerol, which is released through lipolysis can also enter the glycolysis pathway, where it can contribute to either energy generation through entry into the tricarboxylic acid (TCA) cycle, or it can enter the pentose phosphate pathway (PPP) and can support inflammation[51]. We therefore looked at the expression of genes associated with glycolysis and the TCA cycle (Supplementary Fig. 9A). We found that genes associated with glycolysis and the downstream generation of lactate (*Hk2*, *Ldha*) were upregulated during infection, across all preadipocyte populations. We also found upregulation of key TCA cycle genes, including those encoding for succinate dehydrogenase (*Sdhb*) and malate dehydrogenase (*Mdh1*, *Mdh2*) during infection, which could contribute to reactive oxygen species (ROS) generation. Finally, we explored genes in the pentose phosphate pathway (Supplementary Fig. 9B) and found upregulation of key genes within this pathway (including *Aldoa*, *Gpi1*). Taken together, these results indicate that during early infection time points, interstitial and committed preadipocytes upregulate gene pathways associated with IL-17 signalling, lipolysis and downstream metabolic pathways.

### IL-17A/F signalling in adipocytes is critical for controlling tissue wasting and local parasite burden in the iWAT

Our scRNAseq dataset predicts that both interstitial and committed preadipocytes express *Il17ra* during *T. brucei* infection, rendering them able to sense local IL-17A/F. Since committed preadipocytes are on a fixed trajectory to a mature adipocyte lineage[52], we asked whether IL-17A/F signalling through adipocytes plays a role in driving changes in bodyweight, iWAT mass and/or iWAT lipid content. To address this, we generated mice with a conditional deficiency of *Il17ra* in white adipocytes, across all white adipose tissue depots (*Adipoq*$^{Cre}$ × *Il17ra*$^{fl/fl}$) and infected them alongside C57BL/6 controls, housing them in single-housing. We observed that adipocytes from *Adipoq*$^{Cre}$ × *Il17ra*$^{fl/fl}$ mice maintained their bodyweight (and at points increased their weight) throughout the course of infection compared to C57BL/6 controls, which lose bodyweight, in particular after 20 dpi (Fig. 8A). These results mirrored the results obtained from the *Il17af*$^{-/-}$ mice, which were also protected from the infection-induced weight loss (Fig. 5D and Supplementary Fig. 10A). In addition, we observed that both *Il17af*$^{-/-}$ and *Adipoq*$^{Cre}$ × *Il17ra*$^{fl/fl}$ mice had a higher food intake over the course of infection compared to C57BL/6 controls (Fig. 8B and Supplementary Figure 10B), suggesting that IL-17A signalling regulates both bodyweight and food intake. Next, we assessed whether adipocyte IL-17A signalling influenced iWAT mass during *T. brucei* infection. As with mice globally deficient for IL-17A/F (Fig. 5F), infected *Adipoq*$^{Cre}$ × *Il17ra*$^{fl/fl}$ mice also experienced less iWAT wasting compared with C57BL/6 mice (Fig. 8C). We also found that in contrast to naive C57BL/6 mice, *Adipoq*$^{Cre}$ × *Il17ra*$^{fl/fl}$ mice had a higher frequency of small adipocytes, particularly in the 100−300 μm$^2$ range (Fig. 8D, E). Infected *Adipoq*$^{Cre}$ × *Il17ra*$^{fl/fl}$ mice also had higher frequencies of small adipocytes in the 51−200 μm$^2$ range, compared with C57BL/6 mice (Fig. 8F). The significantly higher frequency of smaller adipocytes in the *Adipoq*$^{Cre}$ × *Il17ra*$^{fl/fl}$ mice may arise from an accumulation of interstitial preadipocytes, as shown in our single-cell dataset (Fig. 7), which are unable to differentiate to mature adipocytes. Together, this suggests that while the infected *Adipoq*$^{Cre}$ × *Il17ra*$^{fl/fl}$ mice retain more iWAT mass, the composition of the tissue is changed compared with C57BL/6 mice. To test this, we measured the expression of *Pi16* and *Dpp4*, which were exclusively upregulated in preadipocytes in our single-cell dataset (Supplementary Fig. 10C), as well as *Pparg* as a marker of mature adipocytes. Although we observed no significant increase in *Dpp4* or *Pi16* expression in the iWAT of infected C57BL/6 mice, both genes were significantly upregulated in the iWAT of infected *Adipoq*$^{Cre}$ × *Il17ra*$^{fl/fl}$

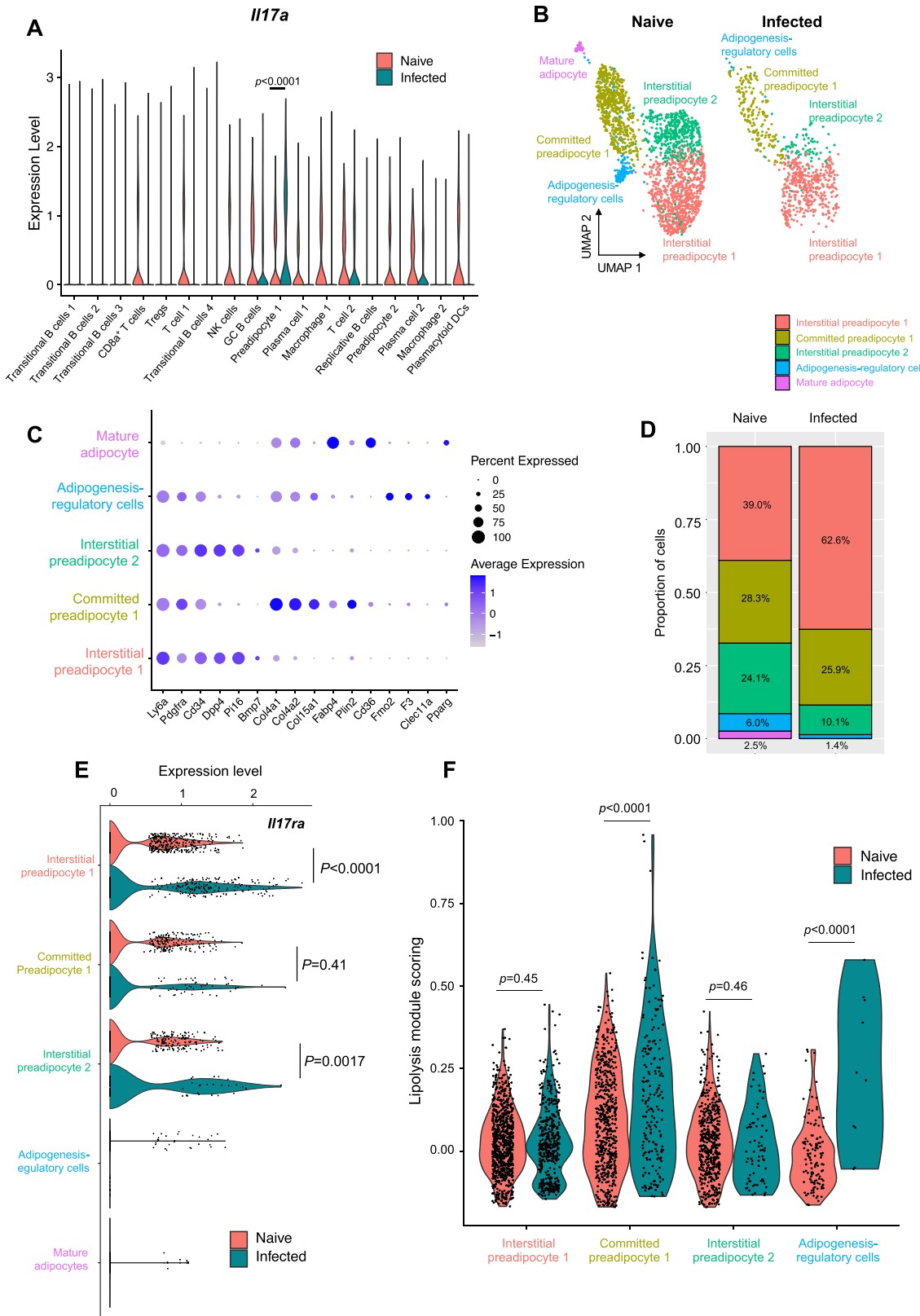

**Fig. 7 | _T. brucei_ infection leads to upregulation of the IL-17A receptor and genes associated with lipolysis in preadipocytes. A** Violin plot showing the expression level of the IL-17A receptor (_Il17ra_) across all cell populations in our scRNAseq dataset. **B** UMAP of subclustered preadipocytes. **C** Dot plot showing expression of genes used to identify specific populations of preadipocytes. **D** Frequency plot of preadipocyte populations. **E** Violin plot showing expression of

_Il17ra_ in specific preadipocyte populations. **F** Lipolysis gene module score of genes typically associated with lipolysis across all preadipocyte subsets detected in (**B**). Single-cell data are comprised of one technical replicate per condition, each containing pooled cells from the iWAT of five male mice. For (**A, E, F**), data were analysed using a one-way ANOVA.

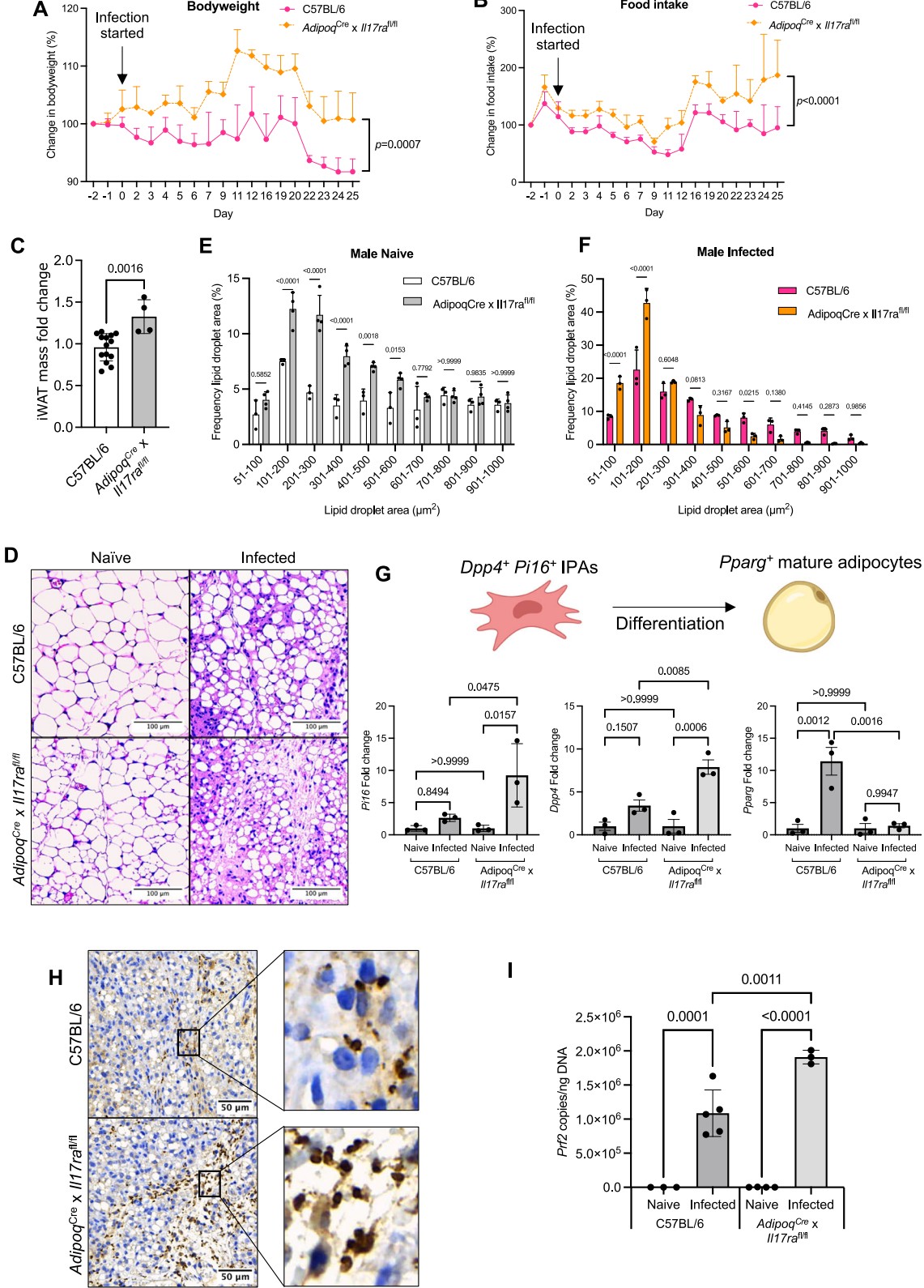

mice (Fig. 8G). In addition, *Pparg* was upregulated in the iWAT of infected C57BL/6 mice, but unchanged in infected *Adipoq^Cre^ × Il17ra^fl/fl^* mice (Fig. 8G). Together, these data indicate that in the absence of functional IL-17 signalling on adipocytes, the *Pi16^+^ Dpp4^+^* interstitial preadipocytes are unable to complete their differentiation and/or developmental programme successfully towards *Pparg+* mature adipocytes (Fig. 8G). Unexpectedly, we also found a significantly

higher number of parasites in the iWAT of *Adipoq^Cre^ × Il17ra^fl/fl^* compared with C57BL/6 mice (Fig. 8H, I), indicating that signalling through the adipocyte IL-17A receptor plays a central role in controlling local parasite numbers. These data demonstrate that IL-17 signalling is critical for promoting and/or supporting preadipocyte development towards mature adipocytes in the context of *T. brucei* infection, as well as impacting local parasite tissue burden.

**Fig. 8 | Adipocyte IL-17A signalling is critical for control of parasite numbers in the iWAT. A** Percentage changes in bodyweight of male C57BL/6 or *Adipoq*^Cre^ × *Il17ra*^Flox^ mice. **B** Percentage changes in food intake of singly housed male C57BL/6 or *Adipoq*^Cre^ × *Il17ra*^Flox^ mice. **C** iWAT mass at 25 days post-infection or in naive mice. iWAT was dissected and weighed before normalising to bodyweight, to account for variation between biological replicates. *N* = 14 (C57BL/6) or 4 (*Adipoq*^Cre^ × *Il17ra*^fl/fl^) biological replicates per group, collected from 1-2 independent experiments. **D** Representative histological H&E staining of iWAT showing adipocyte lipid droplets and immune infiltrate. Analysis of lipid droplet area (μm²) in naive (**E**) and infected (**F**) males. *N* = 3 biological replicates per group from one experiment. Lipid droplets were measured from three distinct areas in each image and then combined for each biological replicate. **G** RT-qPCR of *Pi16*, *Dpp4*, and *Pparg* from the iWAT.

Cartoon illustrates gene expression by interstitial preadipocytes (IPAs) or mature adipocytes. *N* = 3 biological replicates per group from one experiment. Cartoon created with Biorender (Agreement number RY25XVY5HT). **H** Histological analysis of the iWAT trypanosome colonisation. **I** Parasite density in the iWAT, which was measured by RT-qPCR of genomic DNA. *N* = 3 (C57BL/6 naive, *Adipoq*^Cre^ × *Il17ra*^fl/fl^ infected), 4 (*Adipoq*^Cre^ × *Il17ra*^fl/fl^ naive) or 5 (C57BL/6 infected) biological replicates per group from one experiment. **A**, **B**, **E**, **F** Data were analysed using a two-way ANOVA with Sidak post hoc testing. **C** Data were analysed using a two-tailed Student's *t* test. **G**, **I** Data were analysed using a one-way ANOVA with Tukey's post hoc testing. Biological replicates are indicated by symbols for histograms. For other plots, *n* = 4 biological replicates/group. Data for all panels are expressed as mean ± SD. IPA interstitial preadipocyte.

## Discussion

It is now clear that African trypanosomes establish dynamic interactions with several tissues, leading to the establishment of infectious foci important for disease transmission and pathogenesis. In this context, the adipose tissue has emerged as a critical site for survival, replication, and antigenic diversity, as recently shown[53]. In this study, we set out to determine the impact that African trypanosomes have on the subcutaneous white adipose tissue, which is adjacent to the skin and might be critical for forward transmission. In our recent work, we identified that skin *Il17a*^+^ Vγ6^+^ cells play an important role in controlling subcutaneous white adipose tissue wasting[54], and here we wanted to explore the role of IL-17A in greater detail, focusing exclusively on the iWAT. Through various complementary analyses, we uncovered a previously unappreciated role of IL-17A/F in controlling tissue dynamics in response to *T. brucei* infection, acting on interstitial and/ or committed preadipocytes to support adipocyte maturation and tissue replenishment under chronic inflammatory challenges. Notably, we also observed elevation of circulating IL-17A in HAT patients, suggesting that this cytokine may play a role in the human immune response to *T. brucei* infection. It is, therefore, tempting to speculate that IL-17A could also play a role in mediating the weight loss experienced by patients infected with *T. brucei*, although this remains to be studied in more detail in the context of natural infections in humans.

Previous studies have focused primarily on the use of a single sex during *T. brucei* infection, with a limited number of studies comparing sexes when measuring effects on reproductive organs[55,56]. It is understood that males and females display multiple differential responses to infection, including differences in sickness behaviour[57], weight loss[58], and the immune response[59]. This aligns with the results presented here, where we observe differences in behaviour, weight loss and the immune response between male and female mice during infection. More specifically, weight loss in male mice was coupled with decreased iWAT mass, and was associated with a decrease in adipocyte lipid droplet size in male compared to female mice, potentially indicating that male mice utilise their lipid stores at a faster rate than females. One explanation for this is the upregulation of *Il17ra* in the iWAT of male, but not female, mice during *T. brucei* infection, which could render males more responsive to IL-17A signalling. Similar effects have been observed in humans with the condition ankylosing spondylitis, which results in upregulation of *IL17RA* in men, but not women[60]. Moreover, studies of periodontal disease using global *Il17ra*-deficient mice found that female mice lacking IL-17RA were more susceptible to bone loss than males[61], indicating that the effects of IL-17A signalling are context- and sex-dependent. During *T. brucei* infection, it is possible that sex hormones, such as testosterone, play a role in promoting higher rates of lipolysis in infected male mice compared with females[62], or decreases in oestrogen production in infected females, which could also inhibit lipolysis. However, further work is required to determine whether sex hormones directly influence iWAT wasting, or the quality of the anti-parasitic immune response locally, during *T. brucei* infection. Alternatively, the effects that we observe may be a result of females preferentially storing lipids in iWAT compared with

males[63], and so throughout infection they are able to store more fatty acids obtained from their diet than males, slowing down lipid droplet shrinkage. Due to this loss of lipid content, we hypothesised that circulating glycerol, which is traditionally used as a proxy for measuring adipocyte lipolysis[64–66], would be elevated during infection. We found evidence suggesting that, during early infection, metabolic pathways such as lipolysis become more active, likely contributing to the changes in bodyweight and iWAT mass that we observed. However, we found that in both HAT patients and chronically infected mice circulating glycerol levels are diminished during trypanosomiasis, which may occur due to the extensive loss of adipocytes and subsequent decreases in lipolysis. Thus, it is tempting to speculate that since lipolysis is a critical regulator of multiple immune compartments, including macrophages[67] and CD4^+^ T cells, that downregulation of this pathway will also impair the immune response to infection. However, a more comprehensive panel of metabolic markers beyond circulating levels of glycerol could provide a more integrative overview of the systemic metabolic responses to infection. In this context, further work is required to identify such metabolic markers that could specifically and selectively reflect the metabolic state of the iWAT during acute and chronic stages of African trypanosomiasis. During inflammatory conditions such as pancreatitis, inflammation increases secretion of lipases that can drive adipose tissue lipolysis[68], and since the pancreas is heavily colonised by *T. brucei*[69] further work will be important to determine the relative contribution of the pancreas to infection-induced iWAT wasting. To further understand the processes driving the adipocyte shrinkage that we observed during infection, we performed bulk transcriptomic analyses on the iWAT. This supported our findings that adipose tissue lipolysis is diminished during chronic infection, showing extensive downregulation of genes associated with mitochondrial metabolism and lipolysis in both male and female mice. This suppression of metabolic pathways has been observed in response to infection with other pathogens, such as *Mycobacterium tuberculosis*[70]. During *M. tuberculosis* infection of the lungs, transcripts encoding enzymes of oxidative phosphorylation and the tricarboxylic acid cycle are downregulated, and there is a shift to glycolysis[70], which may support the inflammatory response to infection[71]. Furthermore, in diseases such as anorexia nervosa, prolonged weight loss and negative energy balance, lipolysis decreases, which is hypothesised to preserve energy that is needed to maintain processes essential to survival[72]. In the context of *T. brucei* infection, prolonged weight loss may result in the adipose tissue shutting down its main metabolic functions during chronic infection (namely lipolysis and mitochondrial metabolism) to preserve energy stores. Alternatively, it is also possible that the observed downregulation of metabolic transcripts in iWAT in response to infection could be caused by the loss of fully differentiated adipocytes, and therefore, underrepresentation of transcripts from these cells.

Previous studies of the adipose tissue immune response to *T. brucei* infection focused exclusively on the gWAT of male mice, and identified an expansion of macrophage, Teff cell, and B cell populations[55]. In agreement with this study, we also observed an expansion of

macrophages and Teff cells. However, we also noted that there was a reduction in the proportion of B cells in the iWAT of male mice, and no changes in females, which could suggest that this is a tissue- and sex-specific response. Using CyTOF, we also observed an expansion of IL-17⁺ Teff cells in both males and females during infection. This is in contrast to our bulk transcriptomic data, which indicated that this expansion would occur in males but not females and could indicate an over-representation of transcripts related to T$_H$17 differentiation in male mice. Although IL-17A⁺ Teff cells may contribute to increases in circulating IL-17A, since virtually all tissues are colonised during *T. brucei* infection[69], we cannot rule out that IL-17-expressing cells, such as mast cells[73] or innate lymphoid cells[74], in other tissues contribute to circulating IL-17A. Our observation of expanded IL-17-producing T cells in the iWAT led us to question whether IL-17 plays a role in controlling iWAT tissue structure and/or function during *T. brucei* infection. Deficiency of IL-17A/F prevented weight loss in *T. brucei*-infected male mice compared to immunocompetent C57BL/6 male mice, but had no effect of body-weight on female mice. When examining the tissue responses to infection at the single-cell level, we identified that *Dpp4⁺ Pi16⁺* interstitial preadipocytes upregulate *Il17ra* in response to *T. brucei* infection, indicating that the capacity of these cells to sense IL-17 signalling is augmented during infection. These adipogenic precursor cells are critical for maintaining a normal supply of fully mature adipocytes under homeostatic conditions[49], but their dynamics during infection remain poorly understood. Using a conditional deletion system to specifically delete *Il17ra* on adipocytes, we were able to identify a significant increase in the expression levels of *Dpp4* and *Pi16*, consistent with an accumulation of immature adipocytes. In addition to this, we found that adipocyte *Il17ra* deletion prevented upregulation of *Pparg*, further suggesting that these mice are less able to develop mature adipocytes during infection. Furthermore, at the histological level, this targeted deletion resulted in a higher proportion of small adipocytes in the Il17ra-deficient mice compared to the C57BL/6 controls, under both naive and infection conditions. This is in agreement with previous studies showing that deficiency of the adipocyte IL-17A receptor leads to reductions in iWAT lipid content under steady-state conditions[75]. In these previous studies, the reductions in lipid content resulted from the induction of thermogenesis in the iWAT, when adipocyte IL-17A signalling was inhibited, leading to increased energy substrate utilisation. Whilst there is evidence to suggest that non-shivering thermogenesis can be activated during infections such as influenza[76], we did not find evidence of thermogenic activation during *T. brucei* infection, which may indicate that IL-17A signalling can also drive loss of iWAT mass by potentially impairing adipocyte maturation, which has been shown to occur in vitro[9].

Unexpectedly, we found that unlike global deficiency of IL-17A/F, deficiency of the adipocyte IL-17A receptor diminished control of iWAT parasite burden, which raises the possibility that adipocytes are a key contributor to the immune response to *T. brucei* infection. This discrepancy may arise since the IL-17A receptor heterodimerises to the IL-17C receptor and, therefore, transduces signals from IL-17A, IL-17F and IL-17C[77], raising the possibility that IL-17C is critical for the control of local parasite numbers. Regardless, it remains to be determined whether loss of control of parasite numbers occurs because IL-17A receptor signalling is driving changes in lipolysis, or because IL-17A receptor signalling is altering the phenotype of adipocytes in a way that is influencing the local immune response. Supporting the former possibility, we recently used scRNAseq-based cell−cell communication analyses to show that adipocytes upregulate a range of factors, including *Cd40*, *Icam1*, *Jag1*, *Tnf* and *Tnfsf18* in response to *T. brucei* infection, which could contribute to T cell activation[49]. Furthermore, in the same study, we found that preadipocytes upregulate expression of inflammatory cytokines and antigen presentation molecules[17]. The inability to control local parasite numbers when adipocyte IL-17A receptor signalling is abrogated suggests that adipocytes are critical

contributors to and coordinators of the local immune response to *T. brucei* infection, either by direct engagement with the immune system (e.g., via T cell activation), through metabolic fuelling (e.g., lipid mobilisation and release), or both. Recent work also found that when adipocytes can no longer engage lipolysis, there is a transient increase in the number of parasites in the gonadal adipose tissue early during infection[78]. The transient nature of this change could suggest that lipolysis is fuelling a specific arm of the immune response to infection that is necessary for parasite control at that time point. Although there is the possibility that lipolysis can limit parasite numbers due to lipotoxicity[78], the parasites also upregulate genes associated with fatty acid oxidation when they reside in the adipose tissue[6], potentially indicating that they are capable of adapting to the lipid-rich environment and may be resistant to lipotoxicity. The findings that we present here show that when local IL-17A receptor signalling is abrogated, the architecture of the iWAT changes to contain more immature pre-adipocytes compared to C57BL/6 controls, impacting the efficiency of the local immune response, and resulting in a higher parasite burden, once more placing adipocyte dynamics at the core of the immunological responses against *T. brucei* in the adipose tissue.

Based on all these observations, we propose a model whereby chronic *T. brucei* infection leads to an increase in circulating IL-17A/F in humans and animals, which act locally in the adipose tissue, by direct signalling through the adipocyte IL-17 receptor, and that this, in turn, acts to coordinate adipocyte maturation, local immune responses, and efficient and timely parasite control. This study provides insights into the metabolic response of the subcutaneous adipose tissue during infection with *T. brucei* and how changes in cellular metabolism influence systemic metabolism. Future work is necessary to understand how adipose tissue dynamics are coupled to immunological responses in the context of chronic infections, as well as to better understand the drivers of the sexually dimorphic responses that we observed in the iWAT during chronic *T. brucei* infection.

## Methods

### Ethics statement

Male and female human serum samples used for IL-17A measurements were collected in Guinea as part of the National Control Program from healthy controls or from patients infected with *T.b. gambiense*. Participants were informed of the study objectives in their own language and signed a written consent form. Participants comprised six males and four females, all over the age of 18 years. Approval for this study was obtained from the Comité Consultative de Déontologie et d'Ethique (CCDE) of the Institut de Recherche pour le Développement (approval number 1–22/04/2013). Due to limited sample numbers, we did not stratify data by sex. Human serum samples used for glycerol measurements were collected as part of the TrypanoGEN Biobank[79], with ethical approval from the Democratic Republic of Congo National Ministry of Public Health (approval number 1/2013). Samples were from healthy controls or from patients infected with *T.b. gambiense*. Samples were used from different regions due to limitations in sample availability. Ethical approval to use all human samples outlined in this study was given by the University of Glasgow (approval number 200120043). All animal experiments were approved by the University of Glasgow Ethical Review Committee and performed in accordance with the home office guidelines, UK Animals (Scientific Procedures) Act, 1986 and EU directive 2010/63/EU. All experiments were conducted under SAPO regulations and UK Home Office project licence number PC8C3B25C and PP4863348 to Dr Jean Rodgers and Professor Annette MacLeod, respectively.

### Mouse generation and infections with *Trypanosoma brucei*

Eight- to ten-week-old Adipoq-cre (JAX, stock 028020) or Il17ratm2.1-Koll/J (JAX, stock 031000) were purchased from The Jackson Laboratory. These mice were crossed to generate Adipoq^Cre × Il17ra^Flox mice,

with an adipocyte-specific deletion of the IL-17A receptor. Six to eight weeks old male or female C57BL/6J (JAX, stock 000664), *Il17af*[-/-] (JAX, stock 034140), or IL-17A GFP reporter mice (JAX, stock 018472) were also purchased. For all experiments, at >10 weeks old, male or female mice were randomly allocated to control or treatment groups by animal unit technical staff. Mice were then inoculated by intra-peritoneal injection with ~2 × 10³ of the *T. brucei brucei* Antat 1.1[E] parasite strain[80]. Parasitaemia was monitored by regular sampling from tail venesection and examined using phase microscopy and the rapid "matching" method[81]. Uninfected mice of the same strain, sex and age served as uninfected controls. All mice were fed ad libitum and kept on a 12 h light−dark cycle. Room temperature was 20–24 °C and humidity ranged from 50 to 70%. All in vivo experiments were concluded at 25 dpi, to model chronic infection in humans.

### RNA purification and bulk RNA sequencing

iWAT was harvested and stored in TRIzol™ (Invitrogen). Total RNA was then purified using an RNeasy Kit (Qiagen) as per the manufacturer's recommendations. The RNA was purified in 30 μL of nuclease-free water (Qiagen), and RNA concentration measured on a NanoDrop™ 2000 (Thermo Fisher Scientific). Samples were shipped to Novogene (Cambridge, UK) to undergo quality control, library preparation and sequencing. RNA integrity was assessed using an RNA Nano 6000 Assay Kit (Agilent Technologies) with a Bioanalyzer 2100 (Agilent Technologies), as per the manufacturer's instructions. Samples with an RNA integrity number (RIN) of >6.0 were qualified for RNA sequencing.

**Library preparation.** Library preparation was performed by Novogene (Cambridge, UK). Messenger RNA (mRNA) was purified from total RNA using poly-T oligo-attached magnetic beads. Fragmentation was carried out using divalent cations under elevated temperature in a First-Strand Synthesis Reaction Buffer (5X). First-strand cDNA was synthesised using random hexamer primers and M-MuLV Reverse Transcriptase (RNase H-). Second-strand cDNA synthesis was then performed using DNA Polymerase I and RNase H. Remaining overhangs were converted to blunt ends via exonuclease/polymerase activity. Following adenylation of 3' ends of DNA fragments, adaptors with hairpin loop structures were ligated. To select cDNA fragments of 370−420 bp in length, library fragments were purified using AMPure XP beads (Beckman Coulter), as per the manufacturer's instructions. PCR was then performed using Phusion High-Fidelity DNA polymerase, Universal PCR primers, and Index (X) primers. Finally, PCR products were purified (AMPure XP system) using AMPure XP beads (Beckman Coulter), as per the manufacturer's instructions, and library quality was assessed using a Bioanalyzer 2100 (Agilent Technologies).

**Sequencing and data analysis.** Clustering of the index-coded samples was performed on a cBot Cluster Generation System using a TruSeq PE Cluster Kit v3-cBot-HS (Illumia) according to the manufacturer's instructions. After cluster generation, libraries were sequenced on an Illumina Novaseq 6000 platform and 150 bp paired-end reads were generated. Raw reads in fastq format were processed through proprietary Perl scripts developed by Novogene (Cambridge, UK). Clean reads were obtained by removing reads containing adaptors, poly-N, or low-quality reads from raw data. Concurrently, the Q20, Q30 and GC content of the clean data was calculated. Genome and genome annotation files (Genome Reference Consortium Mouse Build; GRCm39) were downloaded. An index of the reference genome was built using Hisat2 (v2.0.5) and paired-end clean reads were aligned to the reference genome using Hisat2 (v2.0.5).

The featureCounts (v. 1.5.0-p3) package was used to count read numbers mapped to each gene, before calculating the Fragments Per Kilobase of transcript sequence per Millions base pairs (FPKM) of each gene using the length of the gene and reads count mapped to this gene.

Differential expression analysis was performed using the DESeq2 R package (v. 1.20.0). The resulting *P* values were adjusted using the Benjamini and Hochberg approach to control false discovery rate (Supplementary Data 2 and 4). Genes with an adjusted *P* value of <0.05 were assigned as differentially expressed. Pathway enrichment analysis of differentially expressed genes was performed using the DAVID platform, mapping genes to the KEGG database. KEGG terms with an adjusted *P* value < 0.05 were considered significantly enriched (Supplementary Data 3). Heatmaps were generated using the pheatmap (Version 1.0.12) and Tidyverse packages in R (Version 4.2.1). Samples were clustered by Euclidean distance. PCA analysis was performed using ggplot2, with naive and infected male and female mice plotted separately.

### Adipose tissue processing and preparation of single-cell suspension for single-cell RNA sequencing

Infected animals and naive controls were anaesthetised with isoflurane at 7 dpi and perfused transcardially with 25−30 mL of ice-cold 1× PBS containing 0.025% (wt/vol) EDTA, and both iWAT pads were excised, and the inguinal lymph node removed. For each condition, fat pads were pooled from five mice, resulting in a single technical replicate per condition. The iWAT was dissociated using an Adipose Tissue Dissociation Kit, mouse and rat (Miltenyi Biotec) with a gentleMACS™ Octo Dissociator with Heaters (Miltenyi Biotec) as per the manufacturer's recommendations. Digested tissue was then passed through a 70-μm and then a 40-μm nylon mesh filter, which were washed with DMEM. The suspension was then centrifuged at 400×*g* at 4 °C for 5 min to isolate the stromal vascular fraction and remove adipocytes. Finally, cells were passed through a MACS dead cell removal kit (Miltenyi Biotec) and diluted to ~1000 cells/μL in 200 μL HBSS 0.04% BSA and kept on ice until single-cell capture using the 10X Chromium platform. The single-cell suspensions were loaded onto independent single channels of a Chromium Controller (10X Genomics) single-cell platform. Single cells were loaded for capture using 10X Chromium NextGEM Single cell 3 Reagent kit v3.1 (10X Genomics). Following capture and lysis, complementary DNA was synthesised and amplified (12 cycles) as per the manufacturer's protocol (10X Genomics). The final library preparation was carried out as recommended by the manufacturer with a total of 14 cycles of amplification. The amplified cDNA was used as input to construct an Illumina sequencing library and sequenced on a Novaseq 6000 sequencers by Glasgow Polyomics.

### Read mapping, data processing and integration

For FASTQ generation and alignments, Illumina basecall files (*.bcl) were converted to FASTQs using bcl2fastq. Gene counts were generated using Cellranger v.6.0.0 pipeline against a combined *Mus musculus* (mm10) and *Trypanosoma brucei* (TREU927) transcriptome reference. After alignment, reads were grouped based on barcode sequences and demultiplexed using the Unique Molecular Identifiers (UMIs). The mouse-specific digital expression matrices (DEMs) from all six samples were processed using the R (v4.2.1) package Seurat v4.1.017. Additional packages used for scRNAseq analysis included dplyr v1.0.7 18, RColorBrewer v1.1.2 (http://colorbrewer.org), ggplot v3.3.5 19, and sctransform v0.3.3 20. We initially captured 50,640 cells mapping specifically against the *M. musculus* genome across all conditions and biological replicates, with an average of 20,209 reads/cell and a median of 1,302 genes/cell (Supplementary Data 4). The number of UMIs was then counted for each gene in each cell to generate the digital expression matrix (DEM). Low-quality cells were identified according to the following criteria and filtered out: (i) nFeature >100 or <5,000, (ii) nCounts >100 or <20,000, (iii) >30% reads mapping to mitochondrial genes, and (iv) >40% reads mapping to ribosomal genes, (v) genes detected <3 cells. After applying this cut-off, we obtained a total of 46,546 high-quality mouse-specific cells with a median of 1296 genes/cell (Supplementary Data 4). High-quality cells

were then normalised using the SCTransform function, regressing out for total UMI and genes counts, cell cycle genes, and highly variable genes identified by both Seurat and Scater packages, followed by data integration using IntegrateData and FindIntegrationAnchors. For this, the number of principal components were chosen using the elbow point in a plot ranking principal components and the percentage of variance explained.

## Cluster analysis, marker gene identification, subclustering and differential expression analyses

The integrated dataset was then analysed using RunUMAP (10 dimensions), followed by FindNeighbors (10 dimensions, reduction = "pca") and FindClusters (resolution = 0.4). With this approach, we identified a total of 16 cell clusters The cluster markers were then found using the FindAllMarkers function (logfc.threshold = 0.25, assay = "RNA"). To identify cell identity confidently, we employed a supervised approach. This required the manual inspection of the marker gene list followed by and assignment of cell identity based on the expression of putative marker genes expressed in the unidentified clusters. To increase the resolution of our clusters to help resolve potential mixed cell populations embedded within a single cluster, we subset preadipocytes and T cells and analysed them separately using the same functions described above. In all cases, upon subsetting, the resulting objects were reprocessed using the functions FindVariableFeatures, ScaleData, RunUMAP, FindNeighbors, and FindClusters with default parameters. The number of dimensions used in each case varied depending on the cell type being analysed but ranged between 5 and 10 dimensions. Cell type-level differential expression analysis between experimental conditions was conducted using the FindMarkers function (min.pct = 0.25, test.use = Wilcox) and (DefaultAssay = "SCT"). Cell–cell interaction analysis mediated by ligand-receptor expression level was conducted using NicheNet[82] with default parameters using "mouse" as a reference organism, comparing differentially expressed genes between experimental conditions (condition_oi = "Infected", condition_reference = "Uninfected"). Differential expression analysis was performed using the DESeq2 R package (v. 1.20.0). The resulting $P$ values were adjusted using the Benjamini and Hochberg approach to control false discovery rate (Supplementary Data 7).

## DNA purification

Tissues were harvested from mice and snap-frozen. Tissue was digested using a DNeasy Blood and Tissue kit (Qiagen), before purifying DNA as per the manufacturer's instructions. DNA was eluted in 100 μL of EB buffer (Qiagen).

## Tissue parasite burden quantification

To quantify *T. brucei* parasites in tissue, we amplified 18S ribosomal DNA genes from the gDNA of a known mass of tissue, using RT-qPCR Brilliant II Probe Master Mix (Agilent Technologies) with a TaqMan™ TAMRA Probe system (Applied Biosystems). Reactions were performed in a 25 μL reaction mix comprising 1X Taqman Brilliant III master mix (Agilent, Stockport, UK), 0.2 pmol/μL forward primer (CCAACCGTGTGTTTCCTCCT), 0.2 pmol/μL reverse primer (CGG CAGTAGTTTGACACCTTTTC), 0.1 pmol/μL probe (FAM- CTTGT CTTCTCCTTTTTTGTCTCTTTCCCCCT-TAMRA) (Eurofins Genomics, Germany) and 20 ng template DNA. A standard curve was constructed using a serial tenfold dilution range: $1 \times 10^6$ to $1 \times 10^1$ copies of PCR 2.1 vector containing the cloned *Pfr2* target sequence (Eurofins Genomics, Germany). The amplification was performed on an ARIAMx system (Agilent, USA) with a thermal profile of 95 °C for 3 min followed by 45 cycles of 95 °C for 5 s and 60 °C for 10 s. The *Pfr2* copy number within each 20 ng DNA sample was calculated from the standard curve using the ARIAMx qPCR software (Agilent, USA) as a proxy for the estimated trypanosome load.

## Real-time quantitative PCR

Following RNA purification, cDNA was synthesised, and the RT-qPCR master mix prepared using a Luna Universal One-Step RT-qPCR Kit (New England Biolabs), as per the manufacturer's instructions, with the following primer sequences: *Dpp4* (forward primer: CACCTCTGATG-GAAGCAGCTTC, reverse primer: GATAATCGCTGGTCAGAGCTTCG), *Pi16* (forward primer: AACTGGCACGAGGAGCATGAGT, reverse primer: GCCAATTCTCTCAGTCTTGCTCC), *Pparg* (forward primer: GTA CTGTCGGTTTCAGAAGTGCC, reverse primer: ATCTCCGCCAACAGC TTCTCCT), and *Actb* (forward primer: CATTGCTGACAGGATGCA-GAAGG, reverse primer: TGCTGGAAGGTGGACAGTGAGG). The amplification was performed on an ARIAMx system (Agilent, USA) with a thermal profile of 95 °C for 3 min followed by 40 cycles of 95 °C for 10 s and 60 °C for 30 s.

## Histological analyses

Tissues were placed into 4% paraformaldehyde (PFA) and fixed overnight at room temperature. PFA-fixed tissues were then embedded in paraffin, sectioned, and stained by the Veterinary Diagnostic Services Facility (University of Glasgow, UK). Sections were Haematoxylin and Eosin (H&E) stained for lipid droplet measurement analysis, or 3'-diaminobenzidine (DAB) stained for heat-shock protein 70 (HSP70) to detect *T. brucei* parasites. The HSP70 antibody was a kind gift from Professor James D. Bangs. Slide imaging was performed by the Veterinary Diagnostic Services Facility (University of Glasgow, UK) using an EasyScan Infinity slide scanner (Motic) at ×20 magnification. To determine lipid droplet sizes in adipose tissue, images were first opened in QuPath (v. 0.3.2)[83], before selecting regions and exporting to Fiji[84]. In Fiji, images were converted to 16-bit format, and we used the Adiposoft plugin to quantify lipid droplet area within different sections.

## Mass cytometry sample processing

Adipose tissue was dissected out and transferred to PBS, before dissociating using an Adipose Tissue Dissociation Kit for Mouse and Rat (Miltenyi Biotec), using a gentleMACS™ Octo Dissociator with Heaters (Miltenyi Biotec), as per the manufacturer's recommendations. After the final recommended centrifugation, the pellet (containing the immune cells) was resuspended in Dubecco's Modified Eagle Medium (DMEM) to a concentration of $1 \times 10^6$ cells/mL. Cells were activated for 3 h in a round-bottom 96-well plate using Cell Activation Cocktail (with Brefeldin A) (BioLegend). Plates were then centrifuged at 300×$g$ for 5 min and the pellets resuspended in 50 μL of Cell-ID™ Cisplatin-195Pt viability reagent (Standard BioTools, 201306), and incubated at room temperature for 2 min. Cells were washed twice in Maxpar® Cell Staining Buffer (Standard BioTools, 201306), and centrifuged at 300×$g$ at room temperature for 5 min. The CD16/CD32 receptors were then blocked by incubating with a 1/50 dilution of TruStain FcX™ (BioLegend) in PBS at room temperature for 15 min. An antibody cocktail was prepared from the Maxpar® Mouse Sp/LN Phenotyping Panel Kit (Standard BioTools, 201306), containing the following antibodies: Ly6G/C [Gr1] (141Pr, clone RB6-8C5, 1/100), CD11c (142Nd, clone N418, 1/100), CD69 (145Nd, clone H1.2F3, 1/100), CD45 (147Sm, clone 30-F11, 1/200), CD11b (148Nd, clone M1/70, 1/100), CD19 (149Sm, clone 6D5, 1/100), CD3ε (152Sm, clone 145-2C11, 1/100), TCRβ (169Tm, clone H57-597, 1/100), CD44 (171Yb, clone IM7, 1/100), CD4 (172Yb, clone RM4-5, 1/100). Additionally, we added the following antibodies (Standard BioTools) to our panel: TCRgd (159Tb, clone GL3, 1/100, 3159012 C), and CD27 (150Nd, clone LG.3A10, 1/100, 3150017B). Cells were incubated with antibodies for 60 min, on ice before washing 3 times in Maxpar® Cell Staining Buffer (Standard BioTools). Following staining, cells were fixed in 2% paraformaldehyde (PFA) overnight at 4 °C. Cells were then washed twice with 1× eBioscience™ Permeabilization Buffer (Invitrogen) at 800×$g$ at room temperature for 5 min. The pellets were resuspended in intracellular antibody cocktail (Standard BioTools):

IL-17A (174Yb, clone TC11-18H10.1, 1/100, 3174002 C) and IFNγ (165Ho, clone XMG1.2, 1/100, 3165003 C). Cells were then incubated at room temperature for 45 min. Cells were washed 3 times in Maxpar® Cell Staining Buffer (Standard BioTools) at 800×g. The cells were then resuspended in 4% PFA at room temperature for 15 min, before collecting the cells at 800×g and resuspending in Cell-ID™ Intercalator-Ir (Standard BioTools). Finally, the cells were barcoded by transferring the stained cells to a fresh tube containing 2 μL of palladium barcode from the Cell-ID™ 20-Plex Pd Barcoding Kit (Standard BioTools). Cells were then frozen in a freezing solution (90% FBS and 10% DMSO), before shipping to the Flow Cytometry Core Facility at the University of Manchester for data acquisition. Samples were acquired using a CyTOF XT (Standard BioTools), and analysed using the Cytobank platform (Beckman Coulter Life Sciences).

### Flow cytometric analyses

Infected animals and naive controls were anaesthetised with isoflurane and perfused transcardially with 25–30 mL of ice-cold 1X PBS containing 0.025% (wt/vol) EDTA. iWAT pads were excised and the inguinal lymph node was removed. The iWAT was dissociated using an Adipose Tissue Dissociation Kit, mouse and rat (Miltenyi Biotec) with a gentleMACS™ Octo Dissociator with Heaters (Miltenyi Biotec) as per the manufacturer's recommendations. Digested tissue was then passed through a 70-μm and then a 40-μm nylon mesh filter, and then washed with DMEM. The suspension was then centrifuged at 400×g at 4 °C for 5 min to isolate the immune cell fraction and remove adipocytes. The resulting suspension was seeded on a 96-well plate and stimulated with 1× Cell Activation Cocktail containing phorbol 12-myristate 13-acetate (PMA), Ionomycin, and Brefeldin A (BioLegend) for 3 h at 37 °C and 5% $CO_2$. For flow cytometry analysis, single-cell suspensions were resuspended in ice-cold FACS buffer (2 mM EDTA, 5 U/mL DNAse I, 25 mM HEPES and 2.5% foetal calf serum (FCS) in 1× PBS). Viability staining was performed using Zombie Green Fixable Viability Dye at 1/1000 dilution (BioLegend), and then samples were stained at 1/400 dilution with the following antibodies from BioLegend: F4/80-PE/Cy7 (clone BM8, 1/400), CD19-PE/Cy7 (clone 6D5, 1/400), TCRγδ-BV421 (clone GL3, 1/400), CD27-APC (clone LG.3A10, 1/400), CD45-PE or CD45-BV421 (clone 30-F11, 1/400), CD3ε-PE/Dazzle594 (clone 145-2C11, 1/400) or CD4-PE (clone RM4-4, 1/400). Samples were run on a flow cytometer LSRFortessa (BD Biosciences) and analysed using FlowJo software version 10 (Treestar).

### Quantification of cytokine titres

To measure cytokine titres in murine serum samples, we used a U-PLEX Biomarker kit (Meso Scale Discovery), as per the manufacturer's instructions. Samples were analysed using a MESO QuickPlex SQ 120 (Meso Scale Discovery). IL-17A titres in human serum samples were quantified using a multiplex cytokine panel (Bio-Plex Pro Human Cytokine Assay, BioRad) and a LuminexCorp Luminex 100 machine as per the manufacturer's instructions.

### Quantification of glycerol concentration

Glycerol concentration in either control or infected human or mouse serum was measured using a Glycerol Assay Kit (Merck, MAK117-1KT) as per the manufacturer's instructions. Data analyses, including the construction of a standard curve, were also performed as per the manufacturer's instructions.

### Statistical analyses

All statistical analyses were performed using Graph Prism Version 8.0 for Windows or macOS, GraphPad Software (La Jolla, CA, USA). The normality of data distribution was measured using the Shapiro–Wilks test. Where indicated, data were analysed by unpaired Student's t test, Mann–Whitney test, one-way analysis of variance (ANOVA) or two-way ANOVA. Data were considered to be significant where $P < 0.05$.

### Reporting summary

Further information on research design is available in the Nature Portfolio Reporting Summary linked to this article.

## Data availability

The GEO accession number for raw bulk transcriptomic sequencing and processed data reported in this paper is GSE210600. The GEO accession number for the raw scRNAseq and processed data reported in this paper is GSE233312. Source data are provided with this paper.

## Code availability

The scripts used to generate single-cell data in this study and the processed data are available at Zenodo (7966849).

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

## Acknowledgements

We firstly thank the TrypanoGEN Network for providing serum samples from patients. We also thank Jean Rodgers for the use of her project licence for performing animal work. We thank the Histology Research Service at Veterinary Diagnostic Services, School of Veterinary Medicine, University of Glasgow. We also thank Nicola Munro, Craig Begley, Scott McCall and Catrina Boyd at the Veterinary Research Facility (University of Glasgow) for maintaining optimal husbandry conditions and comfort for the animals used in this study. We thank Julie Galbraith and Pawel Herzyk (Glasgow Polyomics, University of Glasgow) for their support with single-cell library preparation and sequencing. Finally, the authors would like to thank the Flow Cytometry Core Facility, University of Manchester, UK, for mass cytometry sample acquisition. This work was funded in part by a Wellcome Trust Institutional Strategic Support Fund award (316917-01 awarded to M.C.S.), a Society for Endocrinology Early Career Grant (316705/0 awarded to M.C.S.), and a Wellcome Centre for Integrative Parasitology FutureScope grant [174811-23] to M.C.S. This work was also funded in part by a Wellcome Trust Senior Research Fellowship (209511/Z/17/Z awarded to A.M.L.). J.F.Q. is funded by a Sir Henry Wellcome postdoctoral fellowship (221640/Z/20/Z awarded to J.F.Q.). P.C. is funded by a Wellcome Centre for Integrative Parasitology FutureScope grant to J.F.Q. (104111/Z/14/Z awarded to J.Q.). G.P.W. is funded by an MRC grant (MR/S009779/1). CB is funded by an MRC grant (MR/W018497/1). NAM is supported by the BBSRC Institute Strategic Programme (BBS/E/D/20002173 and BB/X010937/1). S.K. is funded by the National Institute of Diabetes and Digestive and Kidney Diseases (DK97441, DK125281 and DK127575) and the Howard Hughes Medical Institute.

## Author contributions

Conceptualisation: M.C.S. and J.F.Q. Methodology: M.C.S., J.F.Q., P.R.G.C., J.O., A.C., P.C., A.G., N.R.K., G.P.W., D.M.N., B.B., M.C., C.B., S.K. and A.M.L. Formal analysis: M.C.S., J.F.Q., P.C., P.R.G.C. and J.O. Writing —original draft: M.C.S. Writing—reviewing and editing: M.C.S., J.F.Q., P.R.G.C., J.O., A.G., G.P.W., C.B., S.K., N.A.M. and A.M.L. Writing—final edits: M.C.S. Funding acquisition: M.C.S., J.F.Q. and A.M.L.

## Competing interests

The authors declare no competing interests.

## Additional information

[1]Wellcome Centre for Integrative Parasitology, University of Glasgow, Glasgow, UK. [2]School of Biodiversity, One Health and Veterinary Medicine, University of Glasgow, Glasgow, UK. [3]Division of Cardiovascular Science, University of Manchester, Manchester, UK. [4]School of Infection and Immunity, University of Glasgow, Glasgow, UK. [5]Department of Parasitology, National Institute of Biomedical Research, Kinshasa, Democratic Republic of Congo. [6]Member of TrypanoGEN, Kinshasa, Democratic Republic of Congo. [7]Institut de Recherche pour le Développement, Unité Mixte de Recherche IRD-CIRAD 177, Campus International de Baillarguet, Montpellier, France. [8]Programme National de Lutte contre la Trypanosomiase Humaine Africaine, Ministère de la Santé, Conakry, Guinea. [9]Division of Endocrinology, Diabetes and Metabolism, Beth Israel Deaconess Medical Center and Harvard Medical School, Boston, MA, USA. [10]Howard Hughes Medical Institute, Chevy Chase, MD, USA. [11]Centre for Cardiovascular Science, University of Edinburgh, Edinburgh EH16 4TJ Scotland, UK. [12]The Roslin Institute and Royal (Dick) School of Veterinary Studies, University of Edinburgh, Edinburgh, UK. [13]Lydia Becker Institute of Immunology and Inflammation, University of Manchester, Manchester, UK. [14]Division of Immunology, Immunity to Infection and Health, Manchester Academic Health Science Centre, University of Manchester, Manchester, UK. ✉e-mail: matthew.sinton@manchester.ac.uk; juan.quintana@manchester.ac.uk

