## [Peer Review File · Nature Communications]

IL-17 signalling is critical for controlling subcutaneous adipose tissue dynamics and parasite burden during chronic murine *Trypanosoma brucei* infectionREVIEWER COMMENTS

Reviewer #1 (Remarks to the Author):

In this manuscript, Sinton et al analyzed the impact of *T. brucei* infection on iWAT and the intersection with Th17 immune responses. Overall, this is a well-designed and well-controlled study providing interesting insight into weight loss mechanisms in sleeping sickness. However, there are a number of points where authors over-state their findings or which require clarification, as follows:

1. The lesser impact on overall gene expression in female mice by PCA (Figure 3A) is inconsistent with the almost-comparable numbers of dysregulated genes in male vs female mice (Figure 3B). Authors should clarify this discrepancy, and whether the fold changes were equivalent between sexes.
2. Figure 3B: authors should clarify whether scaling was performed separately by sex. Common scaling for male and female mice may help address the discrepancy I raise above.
3. Line 235: authors should verify their gene annotations. *Acat1* and *Acat2* can also participate in lipid metabolism. *Acox1* is a key component of peroxisomal beta-oxidation
4. Lines 241-245 and 252-255: Gene-level data supporting these statements should be provided. Table S2 is insufficient since it only reports statistics aggregated at pathway level
5. There is a contradiction between the sex-specific increase in Th17 transcripts, vs the increase in Th17 cells in both sexes. If this is linked to the differences in IL17A receptor noted by the authors, this should be explicitly discussed by the authors.
6. Lines 279-280: "and may be associated with the induction of a hypometabolic state." This statement is not supported by the data in figure 4, since no link to metabolism is displayed, and overall responses are comparable in male and female mice.
7. Line 282-283: this statement is misleadingly restrictive. Higher levels of Th17 cells and higher IL17 production are detected in both sexes in Figures 4I and 4J.
8. Lines 288-290: "which may suggest that IL-17A/F plays a role in controlling systemic parasite numbers during chronic infection.". This statement is not supported by the data, since differences between WT and KO mice were non-significant.
9. Line 296: "Unlike bloodstream parasite numbers". This statement should be clarified to demonstrate that it only applies to females, since the authors state no significant differences in bloodstream parasites in males.
10. Lines 386-389: While I agree that upregulation of a receptor can represent responsiveness to a given signal, high pre-existing expression of the receptor could also demonstrate responsiveness. This statement should be tempered.
11. Lines 432-433: "Our scRNAseq dataset clearly demonstrates that both interstitial and committed preadipocytes upregulate *Il17ra* expression in response to *T. brucei* infection". Data does not reach statistical significance for committed pre-adipocytes, so this statement should be amended. Also applies to lines 542-543.

Minor issues:

1. Line 225: I'm assuming the authors mean an absolute log₂ fold change >0.5. Please clarify.
2. Figure S2: clearly specify that N1-N4 samples are naïve (uninfected) and I1-I4 samples are infected.

Reviewer #2 (Remarks to the Author):

It has been known for some time that during an experimental infection with *Trypanosoma brucei* animals lose weight, a process that seems associated with the loss of gonadal white adipose tissue mass. In this interesting work, the authors show evidence of a sexually dimorphic response during a trypanosome infection. Unlike females, male mice clearly show a loss in adipose tissue mass, altered tissue function and feeding behavior changes. Furthermore, the authors observed an increase in the presence of TH17 T cells in white adipocyte tissue, which correlates with an increase in the levels of circulating IL-17A/F. IL-17 appears to modulate the sexually dimorphic response as it suppresses food intake in male mice, but has no effect in females. These findings were also confirmed in IL-17 KO mice, which shows no weight loss during a trypanosome infection. A possible IL-17 signal was backed up by transcriptomics analysis. Interestingly, the authors also report on elevated of serum levels of IL-17A in HAT patients, which suggest a possible similar role for this cytokine in humans. Collectively, this is a nicely written story that has significance extending beyond the field of host-pathogens interactions as it seems also relevant in mammalian lipid metabolism and neuroimmunology. I don't think additional experiments are necessary, but I'd like to point out a couple of things.

1. Detecting higher serum levels of IL-17 in HAT patients does not necessarily mean that this cytokine has the same effect during a human infection. In fact, I couldn't find a more detailed clinical description on these small cohort of patients, including aspects like disease stage, infecting trypanosome species, weight loss, and gender, which is important to report based on the observation in animals. Therefore, this finding, although interesting in humans, it is just a good correlation to what the authors observed in infected mice, but it doesn't tell us that IL-17 has a similar role during a HAT infection.

2. The authors should be more cautious about the finding of the reduced levels of glycerol in serum as a proxy for lipolysis in either infected animals or humans. I wonder if other metabolic markers, also indicative of adipose function, were also determined in infected animals. I'm not asking the authors to perform a sex comparative metabolomic analysis of mice serum, but if the data is available, it might worth including it.

Reviewer #3 (Remarks to the Author):

This is a very original work in the field of trypanosomiasis. The scientific community will surely benefit from this new and valuable information related to disease pathology and the role of IL-17 in adipose tissue modulation during trypanosome infection. In addition, the work is showing differences in responses between male and female mice, often overlooked. The authors focus on changes occurring in iWAT, using a large variety of techniques such as bulk RNA-seq, scRNA-seq, validation of protein expression, and the use of knockout mice. Their experimental methodology is sound and delivers an incredible wealth of information. The comments below are aimed at increasing the quality of the manuscript. As such, I hope that the authors will find them helpful and constructive. The cell annotation requires further strengthening to visualize cell populations expressing IL17a. The study is focused on IL-17 signaling in controlling adipose tissue during infections, however IL-17 and adipose tissue are strongly regulated by e.g sex hormones (estrogen and testosterone), and this systemic aspect could be further explored.

- 1) Trypanosomiasis is accompanied by hepatomegaly contributing to changes in body weight. As the liver is the main controller of a lipid metabolism, recording possible infection-induced alterations seems to be relevant for this study. The authors concluded that the

spleen mass increased similarly in both male and female mice. Do authors have similar data related to hepatomegaly and its impact on the overall body weight? What is the impact of the liver on lipid metabolism during infection?

2) Trypanosomiasis was also reported to result in decline of a muscle mass due to progressing catabolic muscle wasting. Does the latter occur in infected female and male mice?

3) Adipose tissue is regulated by hormones (e.g., insulin) and sex hormones: estrogen, testosterone, progesterone, and gonadotropin-follicle stimulating hormone. In addition, the production of IL-17 was also shown to be regulated by estrogen and testosterone. The authors study changes between body weight/adipose tissue during trypanosomiasis in female and male mice. In this context - What are the systemic levels of IL-17 and sex hormones during the course of an infection? The study is very focused on iWAT, however higher-level systemic regulatory mechanisms (liver, pancreas, sex hormones) should be considered and discussed.

4) Could authors comment on systemic sources (other organs) of IL-17?

5) The authors conducted cell annotation using scRNA-seq data from 7dpi mice. Some cell populations require further in-depth analysis.

a) Fig 6 C and E. There are 4 subpopulations of Transitional B cells (TransB) annotated. According to the literature, 3 subpopulations of TransBs were described based on flow cytometry analysis, transcriptomics, and their functionality. Could authors provide an explanation in relation to the number of clusters (4) they identified? Which marker genes are behind this clustering? It is not clear from the dot plot? How were the optimal number of clusters chosen?

b) The authors also annotate 2 subpopulations of PCs? Which subpopulations do they represent? Wider selection of canonical markers could help in further annotation.

c) Can authors provide a reference for replicative B cells and T cells? As authors conduct supervised annotation, providing references for used canonical markers should be included.

d) Regarding annotation of T cells and NK cells, it can be difficult to properly annotate NK and NKT cells. In Fig. 6C, annotated NK cells seem to express *Trbc2* (T cell receptor), however NK cells lack a TCR.

e) Some T cells are annotated: T cells 1 and T cells 2. Could authors provide a widely used nomenclature to describe these subpopulations, based on a larger selection of canonical markers?

f) From Figure 6 E, the annotation of Th17 cells requires further strengthening. The analysis could benefit from the use of larger collection of known markers for annotation. How do the authors explain the absence of Th17 from transcriptomic annotation? There is also very low average expression in $V\gamma 6+$ cells, and by very few cells.

g) Macrophage subpopulations are annotated macrophage 1 and 2. Which cell populations do they represent (are these M1/M2)? Myeloid cells can be a source of IL-17, and this cell population expands during infection. Do these cells have elevated expression of *Il17a*?

h) Fig. 6 F and G shows flow cytometry data related to frequency of CD4+Th17 in naïve and infected mice using IL-17A GFP reporter mice. Was this method also used for 25dpi in addition to an intracellular staining Fig.4 I? The frequency seems to significantly differ between 25dpi and 7dpi. Is there a difference in gating strategy? If available, 25dpi scRNA-seq data seem to be more relevant in terms of expansion of IL-17A+ T effector cells.

i) What are the numbers of barcoded cells present in various cell clusters (Fig.6). How does the available cell number impact cell annotation?

6) ScRNA-seq data are often used for DEG analysis, comparing for example different T cells/NK cells between naïve and infected mice. This, in turn, can aid in identification of key significantly expressed genes. It could be demonstrated using a volcano plot. Is *Il17a* and corresponding receptor among DEGs. What are the other DEGs?

7) The scRNA-seq analysis was performed at 7dpi, however the effect of IL-17A on body weight and parasitemia in males can be seen only during chronic infection. Once again - Do authors have scRNA-seq dataset from chronic infection (25 dpi) for comparison and in-depth analysis?

8) Using scRNA-seq data the authors could consider exploring cell-cell interactions between transcriptomic profile of Tregs and Th17/Vgamma6+ cells, taking into account that Treg/Th17 balance is involved in regulation of inflammation. The cell-cell interactions between Il17a expressing cells and preadipocytes could be investigated as well.

Nature Communications Manuscript NCOMMS-23-23052

We thank all of the reviewers for their helpful and constructive comments on our manuscript, and we hope that our revisions satisfy their concerns. Below, we have provided point-by-point answers to the comments made by the reviewers (in blue).

Reviewer #1

In this manuscript, Sinton et al analyzed the impact of *T. brucei* infection on iWAT and the intersection with Th17 immune responses. Overall, this is a well-designed and well-controlled study providing interesting insight into weight loss mechanisms in sleeping sickness.

Authors: We sincerely thank this reviewer for their positive assessment of our work and for taking the time to provide thorough and useful feedback. We believe the revised version of our manuscript is much improved.

However, there are a number of points where authors over-state their findings or which require clarification, as follows:

1. The lesser impact on overall gene expression in female mice by PCA (Figure 3A) is inconsistent with the almost-comparable numbers of dysregulated genes in male vs female mice (Figure 3B). Authors should clarify this discrepancy, and whether the fold changes were equivalent between sexes.

Authors: We thank the reviewer for pointing out this discrepancy. To address this, we generated new PCA plots, with each sex on its own plot, as we believe this approach provides a better resolution of the responses observed in males and females. By doing this, we can now see that levels of variance between naïve and infected mice appear similar when comparing sexes. We have also updated the text to reflect this:

Line 199 *“Principal component analysis (PCA) revealed high levels of variance between naïve and infected male and female mice (Figure 3A).”*

And

Line 879 *“PCA analysis was performed using ggplot2, with naïve and infected male and female mice plotted separately.”*

2. Figure 3B: authors should clarify whether scaling was performed separately by sex. Common scaling for male and female mice may help address the discrepancy I raise above.

Authors: Scaling on the original PCA plot was performed using common scaling. However, we have now presented separate PCA plots for which scaling was performed separately by sex. As mentioned above, we have now generated new PCA plots, with each sex on its own plot, which addresses the discrepancy raised above. The heatmaps now more closely mirror the PCA plots and address the discrepancy highlighted in this comment. We have also added the following text to the methods section:

Line 879 *“PCA analysis was performed using ggplot2, with naïve and infected male and female mice plotted separately.”*

3. Line 235: authors should verify their gene annotations. Acat1 and Acat2 can also participate in lipid metabolism. Acox1 is a key component of peroxisomal beta-oxidation

Authors: We have now updated this text to clarify that genes noted here were examples taken from specific KEGG pathways that we found to be enriched. We have now provided details of which KEGG pathways we are referring to and amended the text as follows:

Line 210 *“Several genes within these pathways related to antigen processing and presentation (KEGG mmu04612; e.g. H2-DMA, H2-DMb1, H2-Ob), cytokine-cytokine receptor interactions (KEGG mmu04060; e.g. Tnf, Ifng, Il1b), and complement and coagulation cascades (KEGG mmu04610; e.g. C2, C3, C4b) (Supplementary Table 2 and 3). When we explored downregulated pathways, we found that the majority of these were related to metabolism (Figure 3E and F), including valine, leucine and isoleucine degradation (KEGG mmu00280; e.g. Acat1, Acat2, Abat) (Supplementary Table 2 and 3) and propanoate metabolism (KEGG mmu00640; e.g. Sucla2, Acox1, Acads) (Supplementary Table 2 and 3). Unexpectedly, we also observed that *T. brucei* infection led to a downregulation of the lipolysis pathway in males and females (KEGG mmu04923) (Supplementary Table 2 and 3), with canonical genes such as Pnpla2, Fabp4, and Lipe being significantly downregulated (Supplementary Figure S2A and S2B).”*

4. Lines 241-245 and 252-255: Gene-level data supporting these statements should be provided. Table S2 is insufficient since it only reports statistics aggregated at pathway level

Authors: We have now updated the text to reference supplementary table 1 and 2, containing details of gene-level and pathway-level expression data, respectively. We have also updated the text as follow:

Line 210 “Several genes within these pathways related to antigen processing and presentation (KEGG mmu04612; e.g. H2-DMA, H2-DMb1, H2-Ob), cytokine-cytokine receptor interactions (KEGG mmu04060; e.g. Tnf, Ifng, Il1b), and complement and coagulation cascades (KEGG mmu04610; e.g. C2, C3, C4b) (**Supplementary Table 2 and 3**). When we explored downregulated pathways, we found that the majority of these were related to metabolism (Figure 3E and F), including valine, leucine and isoleucine degradation (KEGG mmu00280; e.g. Acat1, Acat2, Abat) (**Supplementary Table 2 and 3**) and propanoate metabolism (KEGG mmu00640; e.g. Sucla2, Acox1, Acads) (**Supplementary Table 2 and 3**). Unexpectedly, we also observed that *T. brucei* infection led to a downregulation of the lipolysis pathway in males and females (KEGG mmu04923) (**Supplementary Table 2 and 3**), with canonical genes such as *Pnpla2*, *Fabp4*, and *Lipe* being significantly downregulated (**Supplementary Figure S2A and S2B**). Since lipolytic genes were downregulated, we questioned whether the iWAT was becoming more thermogenic, which could, in turn, lead to a decrease in mass. However, when exploring our data, we also found that canonical genes associated with thermogenesis, such as *Ucp1*, were downregulated in infected males and females (**Supplementary Table 2 and 3**). Moreover, expression of genes associated with UCP1-independent thermogenesis, including *Atp2a1*, *Atp2a2*, and *Atp2a3* were not altered during infection (**Supplementary Table 2 and 3**), suggesting that thermogenesis is not activated in this context.”

5. There is a contradiction between the sex-specific increase in Th17 transcripts, vs the increase in Th17 cells in both sexes. If this is linked to the differences in IL17A receptor noted by the authors, this should be explicitly discussed by the authors.

Authors: This is indeed a puzzling observation. When looking at the pathway-level analysis, we detected a clear enrichment for T_H17-associated transcripts in males but not in females, as shown in figures 3C, D, and G. This apparent lack of T_H17-related transcript enrichment in females could be associated with an overrepresentation of other functional pathways (**Supplementary Table 3**) dominating the tissue transcriptome, but we do not have ways to support this, so it remains a hypothesis. As this reviewer states, we believe that the functional differences observed in the *Il17a/f* KO male mice but not in females, presented in **Figure 5 and S3**, and throughout the other IL-17 related genetic models presented in **Figure 8**, could be associated with the differences in expression level of the IL-17 receptor subunits between sexes, rendering males more “sensitive” to IL-17 signalling than female mice. However, this remains speculative, and now that we have more mechanistic insights into the IL-17 signalling on adipose tissue during infection, we will shift our attention to characterising the sex differences in more detail in future work. We have also included additional text explicitly stating our finding that the IL-17A receptor is upregulated in the iWAT of male but not female mice:

Line 261 “Although we observed an increase in the frequency of IL-17A⁺ Teff cells and circulating IL-17A in both males and females, we only observed an upregulation of the cognate IL-17A receptor (*Il17ra*) in the iWAT of male mice during infection (**Figure 4L**). This could suggest that the iWAT becomes more responsive to IL-17A signalling in males than females during infection.”

We have also included the following additional text in the discussion:

Line 526 “Similar effects have been observed in humans with the condition ankylosing spondylitis, which results in upregulation of IL17RA in men, but not women³. Moreover, studies of periodontal disease using global *Il17ra*-deficient mice found that female mice lacking IL-17RA were more susceptible to bone loss than males⁴, indicating that the effects of IL-17A signalling are context- and sex-dependent.”

Line 577 “Using CyTOF, we also observed an expansion of IL-17⁺ Teff cells in both males and females during infection. This is in contrast to our bulk transcriptomic data, which indicated that this expansion would occur in males but not females and could indicate an overrepresentation of transcripts related to T_H17 differentiation in male mice. Although IL-17A⁺ Teff cells may contribute to increases in circulating IL-17A, since virtually all tissues are colonised during *T. brucei* infection⁵, we cannot rule out that IL-17-expressing cells, such as mast cells⁶ or innate lymphoid cells⁷, in other tissues contribute to circulating IL-17A.”

6. Lines 279-280: “and may be associated with the induction of a hypometabolic state.” This statement is not supported by the data in figure 4, since no link to metabolism is displayed, and overall responses are comparable in male and female mice.

Authors: We have now amended the text to temper our conclusions, as follow:

Line 195 “*T. brucei* iWAT colonisation results in a transcriptional profile indicative of energy conservation in iWAT”

Line 227 *“The downregulation of genes associated with energy metabolism, such as lipolysis and thermogenesis, together with an overall decrease in circulating glycerol levels in serum, could suggest that chronic T. brucei infection leads to mice (and likely in humans) entering a state of energy conservation⁸”*

Line 261 *“Although we observed an increase in the frequency of IL-17A⁺ T_H17 cells and circulating IL-17A in both males and females, we only observed an upregulation of the cognate IL-17A receptor (Il17ra) in the iWAT of male mice during infection (Figure 4L). This could suggest that the iWAT becomes more responsive to IL-17A signalling in males than females during infection. Taken together, our data demonstrate a significant expansion of IL-17-producing T cells in the iWAT in response to T. brucei infection. Moreover, our data could suggest that the iWAT of male mice becomes more responsive to IL-17A signalling during T. brucei infection, compared with females.”*

7. Line 282-283: this statement is misleadingly restrictive. Higher levels of Th17 cells and higher IL17 production are detected in both sexes in Figures 4I and 4J.

Authors: We have amended this sentence to remove references to the sex of the mice, since we observed expansion of T_H17 cells and increased IL-17A in both sexes. We have added the following text:

Line 261 *“Although we observed an increase in the frequency of IL-17A⁺ T_H17 cells and circulating IL-17A in both males and females, we only observed an upregulation of the cognate IL-17A receptor (Il17ra) in the iWAT of male mice during infection (Figure 4L). This could suggest that the iWAT becomes more responsive to IL-17 signalling in males than females during infection.”*

Line 266 *“Moreover, our data could suggest that the iWAT of male mice becomes more responsive to IL-17A signalling during T. brucei infection, compared with females.”*

8. Lines 288-290: “which may suggest that IL-17A/F plays a role in controlling systemic parasite numbers during chronic infection.”. This statement is not supported by the data, since differences between WT and KO mice were non-significant.

Authors: We have now amended this sentence to reflect the fact that we did not observe significant changes in parasite numbers as follow

Line 274 *“Systemically, we found that there were no significant differences in parasitaemia when comparing wildtype and Il17af^{-/-} male (Figure 5A).”*

9. Line 296: “Unlike bloodstream parasite numbers”. This statement should be clarified to demonstrate that it only applies to females, since the authors state no significant differences in bloodstream parasites in males.

Authors: We have now amended this sentence to clarify that bloodstream parasites refers to female mice and not males as follows:

Line 281 *“The parasite burden of the major adipose tissue depots does not appear to be influenced by IL-17A/F in either sex (Figure 5C and Supplementary Figure S3C and S3D), indicating that IL-17A/F is dispensable for controlling local parasite burden in the adipose tissue. We only detected significant differences in the dynamics of circulating parasites in Il17af^{-/-} female mice, which displayed a reduced initial peak of parasitaemia at 5dpi, and a delayed second peak of parasitaemia at 15dpi compared to wildtype controls (Supplementary Figure S3A).”*

10. Lines 386-389: While I agree that upregulation of a receptor can represent responsiveness to a given signal, high pre-existing expression of the receptor could also demonstrate responsiveness. This statement should be tempered.

Authors: We have now amended the text to temper it as follows:

Line 413 *“To this end, we measured IL-17A receptor (Il17ra) expression across all clusters and found that it was exclusively upregulated in preadipocytes during infection (Figure 7A). This may indicate that of all the cells within the iWAT stroma, preadipocytes increase their responsiveness to IL-17A during infection.”*

11. Lines 432-433: “Our scRNAseq dataset clearly demonstrates that both interstitial and committed preadipocytes upregulate Il17ra expression in response to T. brucei infection”. Data does not reach statistical significance for committed pre-adipocytes, so this statement should be amended. Also applies to lines 542-543.

Authors: Although there is a trend in the Il17ra expression in committed and interstitial preadipocyte populations, as shown in figure 7E, only interstitial preadipocytes (clusters 1 and 2) reached statistical significance. Thus, we have now amended the text to reflect that committed preadipocytes do not significantly upregulate Il17ra during infection, as follows:

Line 459 *“Our scRNAseq dataset predicts that both interstitial and committed preadipocytes express Il17ra during T. brucei infection, rendering them able to sense local IL-17A/F.”*

And again in

Line 588 *“When examining the tissue responses to infection at the single cell level, we identified that Dpp4⁺ Pi16⁺ interstitial preadipocytes upregulate Il17ra in response to T. brucei infection, indicating that the capacity of these cells to sense IL-17 signalling is augmented during infection.”*

Minor issues:

1. Line 225: I'm assuming the authors mean an absolute log₂ fold change >0.5. Please clarify.

Authors: We have now updated the text to clarify this point:

Line 203 *“In contrast, compared with naïve controls, this analysis revealed downregulation of 3,332 and 2,606 genes an absolute log₂ fold change >0.5 and a padj of <0.01 in infected male and female mice, respectively (Figure 3B; Supplementary Table 2).”*

2. Figure S2: clearly specify that N1-N4 samples are naïve (uninfected) and I1-I4 samples are infected.

Authors: We have now updated the figure caption to clarify this point:

Line 762 *“Labels N1-N4 refer to naïve samples. Labels I1-I4 refer to infected samples.”*

Reviewer #2

It has been known for some time that during an experimental infection with *Trypanosoma brucei* animals lose weight, a process that seems associated with the loss of gonadal white adipose tissue mass. In this interesting work, the authors show evidence of a sexually dimorphic response during a trypanosome infection. Unlike females, male mice clearly show a loss in adipose tissue mass, altered tissue function and feeding behavior changes. Furthermore, the authors observed an increase in the presence of TH17 T cells in white adipocyte tissue, which correlates with an increase in the levels of circulating IL-17A/F. IL-17 appears to modulate the sexually dimorphic response as it suppresses food intake in male mice, but has no effect in females. These findings were also confirmed in IL-17 KO mice, which shows no weight loss during a trypanosome infection. A possible IL-17 signal was backed up by transcriptomics analysis. Interestingly, the authors also report on elevated of serum levels of IL-17A in HAT patients, which suggest a possible similar role for this cytokine in humans. Collectively, this is a nicely written story that has significance extending beyond the field of host-pathogens interactions as it seems also relevant in mammalian lipid metabolism and neuroimmunology. I don't think additional experiments are necessary, but I'd like to point out a couple of things.

Authors: We are very grateful to this reviewer for such positive and constructive assessment of the work presented in this manuscript. The points raised by this reviewer are important and we have taken them on board as discussed below.

1. Detecting higher serum levels of IL-17 in HAT patients does not necessarily mean that this cytokine has the same effect during a human infection. In fact, I couldn't find a more detailed clinical description on these small cohort of patients, including aspects like disease stage, infecting trypanosome species, weight loss, and gender, which is important to report based on the observation in animals. Therefore, this finding, although interesting in humans, it is just a good correlation to what the authors observed in infected mice, but it doesn't tell us that IL-17 has a similar role during a HAT infection.

Authors: Indeed, we agree with this reviewer and the comparisons between the murine model of infection and the human data should be drawn more carefully. However, we do think it is interesting that the elevation of circulating IL-17 signalling in humans mirror the dynamics observed during experimental infections. We have amended the text to temper any potential overinterpretation obtained from the human data as follow:

Line 257 *“Interestingly, in addition to TNF α and IFN γ , IL-17A was also elevated in the serum of infected mice and compared to naïve controls (Figure 4J), and in first and second stage HAT patients compared to healthy controls (Figure 4K, Supplementary Table 1), suggesting that IL-17A elevation is conserved in mice and humans infected with T. brucei.”*

And again in

Line 510 *“Notably, we also observed elevation of circulating IL-17A in HAT patients, suggesting that this cytokine may play a role in the human immune response to T. brucei infection. It is, therefore, tempting to speculate that IL-17A could also play a role in mediating the weight loss experienced by patients infected with T. brucei, although this remains to be studied in more detail in the context of natural infections in humans.”*

We have also included the available metadata for the cohorts of patients that were profiled for this study (**Supplementary Table 1**), including sample ID, sex, age and location, which we refer to in **Line 188**. We were unable to include details of the ages of the patients that IL-17A measurements were taken from.

2. The authors should be more cautious about the finding of the reduced levels of glycerol in serum as a proxy for lipolysis in either infected animals or humans. I wonder if other metabolic markers, also indicative of adipose function, were also determined in infected animals. I'm not asking the authors to perform a sex comparative metabolomic analysis of mice serum, but if the data is available, it might worth including it.

Authors: We thank this reviewer for highlighting this. Serum glycerol levels are commonly used in the field as a proxy for adipose tissue lipolysis⁹⁻¹¹, although we agree that the field should move towards using a more exhaustive panel to understand adipose tissue metabolic dynamics. Unfortunately, at this stage, we do not have additional data and we anticipate that a more thorough study would be required to identify reliable markers to specifically assess adipose tissue wasting, especially in the context of chronic infections. We have included some additional text in the discussion section to reflect this:

Line 538 *"Due to this loss of lipid content, we hypothesised that circulating glycerol, which is traditionally used as a proxy for measuring adipocyte lipolysis⁹⁻¹¹, would be elevated during infection. We found evidence suggesting that, during early infection, metabolic pathways such as lipolysis become more active, likely contributing to the changes in bodyweight and iWAT mass that we observed. However, we found that in both HAT patients and chronically infected mice circulating glycerol levels are diminished during trypanosomiasis, which may occur due to the extensive loss of adipocytes and subsequent decreases in lipolysis. Thus, it is tempting to speculate that since lipolysis is a critical regulator of multiple immune compartments, including macrophages¹² and CD4⁺ T cells, that downregulation of this pathway will also impair the immune response to infection. However, a more comprehensive panel of metabolic markers beyond circulating levels of glycerol could provide a more integrative overview of the systemic metabolic responses to infection. In this context, further work is required to identify such metabolic markers that could specifically and selectively reflect the metabolic state of the iWAT during acute and chronic stages of African trypanosomiasis."*

Reviewer #3

This is a very original work in the field of trypanosomiasis. The scientific community will surely benefit from this new and valuable information related to disease pathology and the role of IL-17 in adipose tissue modulation during trypanosome infection. In addition, the work is showing differences in responses between male and female mice, often overlooked. The authors focus on changes occurring in iWAT, using a large variety of techniques such as bulk RNA-seq, scRNA-seq, validation of protein expression, and the use of knockout mice. Their experimental methodology is sound and delivers an incredible wealth of information. The comments below are aimed at increasing the quality of the manuscript. As such, I hope that the authors will find them helpful and constructive. The cell annotation requires further strengthening to visualize cell populations expressing IL17a. The study is focused on IL-17 signaling in controlling adipose tissue during infections, however IL-17 and adipose tissue are strongly regulated by e.g sex hormones (estrogen and testosterone), and this systemic aspect could be further explored.

Authors: We thank the reviewer for their positive and constructive response to our manuscript, and for providing feedback that has improved the quality of this work. The main focus of our work was to understand how the subcutaneous adipose tissue responds to *T. brucei* infection, and the rationale for focusing on this adipose tissue depot was that it is in proximity to the skin, making it potentially important for onward transmission of the parasite. We found that there were reductions in iWAT mass in male mice during infection, which could contribute to the observed weight loss in these animals, and this led us to question what could be driving the iWAT wasting. As such, exploring the effects of other organs, such as the liver and muscle, are beyond the scope of this article and we plan to explore these in greater detail in ongoing and future studies. In addition to this, we reported sex differences as these are so commonly ignored, but ultimately decided to focus on the effects of infection on male mice since this was where we saw the strongest phenotype and found evidence that IL-17 drives the effects that we see. Given that the role of IL-17 signalling on adipose tissue dynamics during infection was unreported when we started this study, we prioritised a more in-depth characterisation in male mice, allowing us to build foundational knowledge upon which we plan to expand on the sex differences in ongoing and future studies, as discussed with reviewer 1. We think that the work required to definitively determine the mechanisms driving these sex differences is beyond the scope of this manuscript but remains an interesting and intriguing question.

1) Trypanosomiasis is accompanied by hepatomegaly contributing to changes in body weight. As the liver is the main controller of a lipid metabolism, recording possible infection-induced alterations seems to be relevant for this study. The authors concluded that the spleen mass increased similarly in both male and female mice. Do authors have similar data related to hepatomegaly and its impact on the overall body weight? What is the impact of the liver on lipid metabolism during infection?

Authors: Whilst we do not have data on the contribution of hepatomegaly to bodyweight or altered lipid metabolism, we have noted that there are not histological differences between the livers of males and females

infected with *T. brucei*, as shown below. It is also important to note that the relative liver weights are accounted for when the iWAT, the main focus of our manuscript, was normalised to whole body weight.

2) Trypanosomiasis was also reported to result in decline of a muscle mass due to progressing catabolic muscle wasting. Does the latter occur in infected female and male mice?

Authors: We did not measure muscle mass in this manuscript, but we have performed additional histological analyses the subcutaneous muscle layer from male and female mice following *T. brucei* infection, at 25 days post-infection, and did not observe statistical differences between sexes in response to infection.

3) Adipose tissue is regulated by hormones (e.g., insulin) and sex hormones: estrogen, testosterone, progesterone, and gonadotropin-follicle stimulating hormone. In addition, the production of IL-17 was also shown to be regulated by estrogen and testosterone. The authors study changes between body weight/adipose tissue during trypanosomiasis in female and male mice. In this context - What are the systemic levels of IL-17 and sex hormones during the course of an infection? The study is very focused on iWAT, however higher-level systemic regulatory mechanisms (liver, pancreas, sex hormones) should be considered and discussed.

Authors: This reviewer raises an important point here, that adipose tissue function and IL-17 expression are both influenced by sex hormones. However, as this reviewer rightly comments, our study focusses on understanding the functional consequences of IL-17 signalling on iWAT as the subcutaneous adipose tissue is critical for skin inflammation (as we have recently reported¹³), and for parasite transmission. As we are sure that this reviewer agrees, dissecting other mechanisms underpinning the sex differences that we observe in enough detail would be extensive, likely resulting on a separate publication. As stated before, for the sake of clarity and focus, we decided to centre our attention to characterising the processes occurring in the iWAT of male mice in as much detail as possible, given that this is where we see the most dramatic differences, and that this aspect of the disease was poorly understood. We have included the following text in the discussion, to acknowledge the role of sex hormones and organs such as the liver and pancreas in altering metabolism and bodyweight during infection:

Line 530 "It is possible that sex hormones, such as testosterone, play a role in promoting higher rates of lipolysis in infected male mice compared with females¹⁴, or decreases in oestrogen production in infected females, which

could also inhibit lipolysis. However, further work is required to determine whether sex hormones directly influence iWAT wasting, or the quality of the anti-parasitic immune response locally, during *T. brucei* infection.”

Line 552 “During inflammatory conditions such as pancreatitis, inflammation increases secretion of lipases that can drive adipose tissue lipolysis¹⁵, and since the pancreas is heavily colonised by *T. brucei*⁵ further work will be important to determine the relative contribution of the pancreas to infection-induced iWAT wasting.”

In terms of systemic levels of IL-17 during infection, we have included data from male and female mice in figure 4J, showing elevation of IL-17 in both sexes. In addition, we show that IL-17 is elevated in early and late stage trypanosome infections in humans (Figure 4K). We have also amended the text as follow:

Line 257 “Interestingly, in addition to $TNF\alpha$ and $IFN\gamma$, IL-17A was also elevated in the serum of infected mice and compared to naïve controls (Figure 4J), and in first and second stage HAT patients compared to healthy controls (Figure 4K, Supplementary Table 1), suggesting that IL-17A elevation is conserved in mice and humans infected with *T. brucei*.”

4) Could authors comment on systemic sources (other organs) of IL-17?

Authors: This is a challenging point to address, as virtually all organs are colonised by trypanosomes during the chronic stage of the infection⁵, raising the possibility that all organs can also experience an influx and/or expansion of IL-17-producing cells, such as T_H17 or $V\gamma6^+$ cells as we have recently shown for the skin during chronic *T. brucei* infection¹⁶. The elevation of IL-17 in the circulation of humans and mice infected with *T. brucei* may suggest that this cytokine is likely derived from multiple sources, and not only the iWAT, and most likely from other cellular sources, such as ILC3s and potentially mast cells or other granulocytes/myeloid cells. We have included this important point in the discussion as follow:

Line 581 “Although IL-17A⁺ T_H17 cells may contribute to increases in circulating IL-17A, since virtually all tissues are colonised during *T. brucei* infection⁵, we cannot rule out that IL-17-expressing cells, such as mast cells⁶ or innate lymphoid cells⁷, in other tissues contribute to circulating IL-17A. Our observation of expanded IL-17 producing T cells in the iWAT led us to question whether IL-17 plays a role in controlling iWAT tissue structure and/or function during *T. brucei* infection.”

5) The authors conducted cell annotation using scRNA-seq data from 7dpi mice. Some cell populations require further in-depth analysis.

Authors: We thank this reviewer for this suggestion. We have addressed each of these aspects in the points below.

a) Fig 6 C and E. There are 4 subpopulations of Transitional B cells (TransB) annotated. According to the literature, 3 subpopulations of TransBs were described based on flow cytometry analysis, transcriptomics, and their functionality. Could authors provide an explanation in relation to the number of clusters (4) they identified? Which marker genes are behind this clustering? It is not clear from the dot plot? How were the optimal number of clusters chosen?

Authors: The optimal number of clusters were chosen using the package Clustree¹⁷, which enables the assessment of clusters relationship statistically at various levels of resolutions. We have now included the Clustree visualisation as a supplementary image (Figure S5A). We applied the same approach to all the single cell analyses presented in this manuscript. We have clarified this in the methods section as follow:

Line 332 “These cells were clustered at a resolution of 0.6 using Clustree analysis (Figure S4A). Cells were then broadly classified into 18 clusters (Figure 6A) based on common markers associated with these clusters.”

Authors: In terms of the B cell diversity in our dataset, we initially subset and analyse the B cell compartment separately. After selecting the best level of resolution (= 0.5), We identified a total of eight B cell related clusters based on the expression of B cell-associated markers such as *Cd79a*, *Cd79b*, and *Cd19*, as shown in the new figure S6B. On the basis of the marker genes identified for each of these B cell subsets, we classified these as:

- **Mature B cells** (clusters 0, 1, 2, 3, and 5) that display high expression levels of *Ms4a4c*, *H2-Aa*, *Sell*, *Ltb*, and *Ighd*,
- **Replicative B cells** (clusters 4, 6, and 8; high expression levels of cell-cycle related genes such as *Mki67*, *Top2a*, *Top1*, and *Pclaf*), including germinal centre-like B cells (cluster 4; based on the expression of *Aicda* and *Pcna*)
- **Plasma cells** (cluster 6 and 8; based on the expression of *Sdc1*, *Cd38*, *Jchain*, and *Ighm*)
- **Class-switched plasma cells** (cluster 8; based on the expression of *Cd19*, *Sdc1*, *Cd38*, *Jchain*, and *Ighg1*)

We have amended the manuscript to incorporate these changes as follow:

Line 353 “To gain further insights into the identity of the B cells that we identified in our dataset, we subclustered and reanalysed all of the B cells clusters at a resolution of 0.5 (**Figure S5A**), revealing 9 clusters. Of these clusters, we identified 6 clusters of Mature B cells (Mature B cell 1-6), based on expression of *Cd79a*, *Cd79b*, *Cd19*, *H2-Aa*, *Sell*, *Ltb*, *Ighd*, and *Ighm*^{18,19} (**Figure S5B**). We also identified replicative germinal centre (GC)-like B cells, which we classified based on expression of high expression of *Mki67*, *Top2a*, *Pclaf*, *Aicda*, and *Pcna*^{18,19} (**Figure S5B**). Finally, we identified two populations of plasma cells: *Ighm*⁺ and *Ighg1*⁺ plasma cells, which we classified based on low expression of *Cd19* and high expression of *Sdc1* and *Jchain*^{18,19} (**Figure S5B**).”

b) The authors also annotate 2 subpopulations of PCs? Which subpopulations do they represent? Wider selection of canonical markers could help in further annotation.

Authors: As mentioned above, after reanalysing the B cell compartment in more detail, we concluded that plasma cell (PC) 1 corresponds to *Ighm*⁺ plasma cells, and PC 2 corresponds to class-switched *Ighg1*⁺ plasma cells. We have amended the manuscript to incorporate these changes as follow:

Line 359 “Finally, we identified two populations of plasma cells: *Ighm*⁺ and *Ighg1*⁺ plasma cells, which we classified based on low expression of *Cd19* and high expression of *Sdc1* and *Jchain*^{18,19} (**Figure S5B**).”

c) Can authors provide a reference for replicative B cells and T cells? As authors conduct supervised annotation, providing references for used canonical markers should be included.

Authors: We based the annotation on canonical B and T cell markers for top level annotation and used additional references to support the prediction of potential replicative lymphocyte populations^{18,19}. We have amended the manuscript to incorporate these changes as follow:

Line 357 “We also identified replicative germinal centre (GC)-like B cells, which we classified based on expression of high expression of *Mki67*, *Top2a*, *Pclaf*, *Aicda*, and *Pcna*^{18,19} (**Figure S5B**).”

Line 377 “This resulted in five T cell subclusters, which consisted mainly of Tregs (*Cd4*, *Icos*, *Foxp3*)²⁰, CD8⁺ T cells (*Cd8a*, *Cd8b1*)²⁰, activated NKT cells (*Nkg7*, *Klr1d1*, *Gzma*, *Gzmb*, *Nr4a1*)²⁰, a cluster of activated and replicative T cells with evidence of TCR engagement (*Top2a*⁺, *Mki67*⁺, *Hist1h1b*⁺, *Nr4a1*)²⁰, and IL-17A⁺ V γ 6⁺ cells (*Tcr γ -C1*⁺, *Rorc*⁺, *Cd163l1*⁺, *Il17a*)²¹”

d) Regarding annotation of T cells and NK cells, it can be difficult to properly annotate NK and NKT cells. In Fig. 6C, annotated NK cells seem to express *Trbc2* (T cell receptor), however NK cells lack a TCR.

Authors: We thank the reviewer for this helpful comment. We have now updated the text to clarify that these are NKT cells, and we have added in the additional markers *Klr1c* and *Klrd1* to the dot plot, presented in figure 6C in the revised version of our manuscript, to further support our annotation, in line with previous data on NKT cells²².

e) Some T cells are annotated: T cells 1 and T cells 2. Could authors provide a widely used nomenclature to describe these subpopulations, based on a larger selection of canonical markers?

Authors: Due to overlapping expression of markers such as *Cd8a* and *Cd4*, we were unable to provide a more precise annotation to these cells during our initial analysis. To resolve this, we reclustered and reanalysed all of the T cell clusters, which enabled us to identify the nature of the T cells within each cluster (Figure 6D). We have also added in the following text to clarify this:

Line 375 “As we were unable to resolve the precise identity of the T cell 1 and T cell 2 clusters, we subset (resolution 0.3) and reanalysed the whole T cell compartment separately (**Figure 6E and 6F**).”

f) From Figure 6 E, the annotation of Th17 cells requires further strengthening. The analysis could benefit from the use of larger collection of known markers for annotation. How do the authors explain the absence of Th17 from transcriptomic annotation? There is also very low average expression in V γ 6⁺ cells, and by very few cells.

Authors: Following this reviewers comment, we reanalysed our single cell dataset to understand, in more detail, which cells express *Il17a* in the iWAT, including T_H17 cells. At top level analysis, we found that two clusters, Tregs and T cell 2, expressed *Il17a* (newly included **Figure 6D**). As a result, we decided to focus on exploring which of these cells were expressing *Il17a*. We reclustered and reanalysed all T cells together (**Figure 6E**) and found that V γ 6⁺ T cells, and to a lesser extend replicative T cells, were the most prominent sources of *Il17a* expression (**Figure 6F**), consistent with our previous work¹³. Since we were unable to determine the identity and nature of the cells within the replicative T cell subcluster, we further reanalysed this subcluster separately resulting in the identification of an additional 7 subclusters of replicative T cells, all of which expand during infection (**Figure S7A**). We identified populations of *Cd69*⁺, *Icos*⁺, and *Cd4*⁺ T cells, as well as NKT cells and Tregs (**Figure S7B**). We also identified *Il17a*⁺ T_H17 and V γ 6⁺ cells (**Figure S7B**). We confirmed expression of

Il17a expression in the T_H17 , $V\gamma6^+$, and, to a lesser extent the Treg clusters (**Figure S7C**). This correlates with our validation data, showing an expansion of T_H17 cells at 7 dpi (**Figure 6H**).

We have also updated the manuscript text to acknowledge this new aspect of our analysis:

Line 377 “This resulted in five T cell subclusters, which consisted mainly of Tregs ($Cd4$, $Icos$, $Foxp3$)²⁰, $CD8^+$ T cells ($Cd8a$, $Cd8b1$)²⁰, activated NKT cells ($Nkg7$, $Klr1d1$, $Gzma$, $Gzmb$, $Nr4a1$)²⁰, a cluster of activated and replicative T cells with evidence of TCR engagement ($Top2a^+$, $Mki67^+$, $Hist1h1b^+$, $Nr4a1$)²⁰, and $IL-17A^+$ $V\gamma6^+$ cells ($Tcr\gamma-C1^+$, $Rorc^+$, $Cd163l1^+$, $Il17a$)²¹ (**Figure 6F**). As we were unable to resolve the identity of the replicative T cell cluster, we reclustered and reanalysed this cluster separately (resolution 0.9), revealing 7 clusters (**Figure S7A and Figure S7B**). We classified these as $Cd8a^+$ T cells ($Cd3e^+$, $Cd3g^+$, $Cd8a^+$, $Cd8b1$)²⁰, $Cd4^+$ T cells 1 ($Cd3e^+$, $Cd3g^+$, $Cd4^+$), NKT cells ($Nkg7$, $Klr1d1$, $Trbc1$), T_H17 cells ($Cd3e^+$, $Cd3g^+$, $Cd4^+$, $Il17a^+$), $Cd4^+$ T cells 2 ($Cd3e^+$, $Cd3g^+$, $Cd4^+$), Tregs ($Cd3e^+$, $Cd3g^+$, $Cd4^+$, $Foxp3^+$), and $Il17a$ -expressing $V\gamma6^+$ cells ($Tcr\gamma-C1^+$, $Rorc^+$, $Cd163l1^+$, $Il17a^+$), consistent with our previous work¹³. T_H17 cells were classified based on upregulation of $Il17a$, which we identified using differential expression analysis comparing cells from naïve and infected samples (**Supplementary Table 7**). Furthermore, we identified that $V\gamma6^+$ cells also upregulated $Il17a$ during infection (**Supplementary Table 7**). We next examined whether there was a cell-cell communication axis between T_H17 cells and other replicative T cells found in the adipose tissue during infection (**Figure S7C**). NicheNet analysis predicted that T_H17 cells produce several ligands that can be sensed by Tregs, including $Il17a$ and $Gas6$ (**Figure S7C**). Interestingly, $Gas6$ interactions with Axl on Tregs has previously been shown to enhance the suppressive effects of Tregs, which could explain why $Il17af^{-/-}$ mice develop more severe clinical scores than wild type mice (**Figure 5B**).”

g) Macrophage subpopulations are annotated macrophage 1 and 2. Which cell populations do they represent (are these M1/M2)? Myeloid cells can be a source of $IL-17$, and this cell population expands during infection. Do these cells have elevated expression of $Il17a$?

Authors: We have now renamed Macrophage 1 and Macrophage 2 to Myeloid 1 and Myeloid 2, respectively and, since we were unable to resolve their identities, we reclustered and reanalysed both myeloid and the pDC subsets (**Figure S6A**). By doing this, we identified populations of macrophages, dendritic cells, and neutrophils (**Figure S6B**). We were unable to fully resolve the identity of the macrophage populations and have classified these as either $Ly6i^+$ or $Mrc1^+$ macrophages (**Figure S6B**). We have also included the following text within the manuscript:

Line 361 “To determine the identity of the myeloid cells in our dataset, we subset and reanalysed the Myeloid 1 and 2 and pCD clusters separately (**Figure S6A**) at a resolution of 0.3, revealing 7 clusters. We classified these as $Ly6i^+$ macrophages ($Cd68^+$, $Ms4a6d^+$, $Ly6i^+$, $Pla2g7$)²³, plasmacytoid DCs (pDCs; $Siglech^+$, $Cox6a2^+$, $Ccr9^+$, $Cd300c$)^{24,25}, mature DCs ($Fscn1^+$, $Cacnb3^+$, $Socs2^+$, $Tbc1d4$)²⁶, conventional DC2 (cDC2; $S100a4^+$, $Gpr183^+$, $H2-Eb1^+$, $Marveld1$)²⁷, $Mrc1^+$ macrophages ($Mrc1^+$, $Folr2^+$, $Pf4^+$, $Fcrls$)²³, neutrophils ($Cxcr2^+$, Hdc^+ , $S100a9^+$, $Mmp9$)²⁸⁻³¹, and conventional DC1 (cDC1; $Xcr1^+$, $Gcsam^+$, $Clec9a^+$, $Tlr3$)²⁷ (**Figure S6B**).”

Authors: We also explored whether $Il17a$ was expressed by any of the myeloid clusters within our dataset. As shown in our newly generated **Figure 6D**, we did not detect $Il17a$ expression in any of the myeloid cell clusters.

h) Fig. 6 F and G shows flow cytometry data related to frequency of $CD4^+$ T_H17 in naïve and infected mice using $IL-17A$ GFP reporter mice. Was this method also used for 25dpi in addition to an intracellular staining Fig.4 I? The frequency seems to significantly differ between 25dpi and 7dpi. Is there a difference in gating strategy? If available, 25dpi scRNA-seq data seem to be more relevant in terms of expansion of $IL-17A^+$ T effector cells.

Authors: We thank the reviewer for picking up on this point. We initially began our study focusing on the end point of infection, when mice had lost bodyweight and iWAT mass. We analysed the immune populations of the iWAT using mass cytometry (**Figure 4A - I**), enabling us to multiplex our analysis and gain as much information as possible from the wasted tissue, which was extremely small. We measured the activation markers $CD44$ and $CD69$ in these cells, and gated on activated $CD4^+$ cells and, therefore, $IL-17$ Teff cells were analysed as the frequency of activated cells (**Figure 4I**). We were then interested in understanding events that precede the iWAT wasting, so we started to explore early changes (7 dpi) in the tissue, to understand whether $IL-17A$ was being expressed and may be an early driver of wasting. To examine this, we performed standard flow cytometry of immune cells from the iWAT of $IL-17A$ GFP reporter mice. Here, we first gated on all $IL-17A^+$ cells and then on $CD3e$ and $CD4$, to identify T_H17 cells (**Figure 6G-H and Figure S4**). The discrepancy in cell frequencies between these two figures are likely due to higher expansion of $IL-17A$ -producing cells at 25 dpi, as well as the use of different technological platforms.

In terms of scRNA-seq, we do not have a scRNAseq data collected at 25dpi. While we would have liked to conduct a parallel experiment at 25dpi, due to financial constraints, we decided to focus on characterising the events leading to tissue wasting, starting at 7 dpi. We plan to perform more in depth single cell analyses in future studies.

i) What are the numbers of barcoded cells present in various cell clusters (Fig.6). How does the available cell number impact cell annotation?

Authors: The annotation of the different cell clusters and subclusters was based on known marker genes, for which we have provided relevant references to previous single cell atlases. We think that it is indeed challenging to address the impact of cell annotation based on the number of barcoded cells without having a set of references with various cell titrations screened in parallel using single cell transcriptomics. In our experience, single cell transcriptomics is robust and powerful enough to identify “rare” cell populations, or those that are relatively lowly abundant, such as $V\gamma 6^+$ cells, as we have recently reported¹⁶. We tried where relevant to validate some of the single cell in silico predictions using alternative approaches such as flow cytometry. We have included the number of barcoded cells detected in each of the clusters in **Supplementary table 6** and reference this in the manuscript in **line 334**.

6) ScRNA-seq data are often used for DEG analysis, comparing for example different T cells/NK cells between naïve and infected mice. This, in turn, can aid in identification of key significantly expressed genes. It could be demonstrated using a volcano plot. Is *Il17a* and corresponding receptor among DEGs. What are the other DEGs?

Authors: We thank the reviewer for their suggestion of performing DEG analysis of our dataset. We have now performed DEG analyses on the different subsets of cells, including T cells/NKT cells (**Supplementary Table 7**) and found that three of these show upregulation of *Il17a* during infection: Tregs, $V\gamma 6^+$ cells, and replicative T cells. We have plotted these as volcano plots as shown below, but felt that a data table would be more informative.

Therefore, we have generated new **Supplementary Table 7** in the revised version of our manuscript, where we report the list of differentially expressed genes within the T cell, myeloid cell and preadipocyte subsets. We have also updated the manuscript with the following text:

Line 387 “*T_H17* cells were classified based on upregulation of *Il17a*, which we identified using differential expression analysis comparing cells from naïve and infected samples (**Supplementary Table 7**). Furthermore, we identified that $V\gamma 6^+$ cells also upregulated *Il17a* during infection (**Supplementary Table 7**).”

Authors: Regarding the IL-17A receptor, we show in Figure 7A that *Il17ra* is upregulated only in preadipocytes, and when we recluster and reanalyse preadipocytes, we find that interstitial preadipocyte 1 and interstitial preadipocyte 2 (Figure 7E in the revised version of our manuscript) significantly upregulate *Il17ra*. We have also updated the manuscript with the following text:

Line 434 “Of the preadipocyte populations that we identified, we noted that *Il17ra* expression was upregulated in both interstitial preadipocyte clusters, and was also expressed in the committed preadipocyte cluster, (**Figure 7E**, **Supplementary Table 7**), but not in adipogenesis-regulatory cells or mature adipocytes, suggesting that both interstitial and committed preadipocytes are able to sense IL-17A/F signalling in response to infection.”

7) The scRNA-seq analysis was performed at 7dpi, however the effect of IL-17A on body weight and parasitemia

in males can be seen only during chronic infection. Once again - Do authors have scRNA-seq dataset from chronic infection (25 dpi) for comparison and in-depth analysis?

Authors: In short, we do not have a scRNAseq data collected at 25dpi. As we mentioned above, due to financial constraints, we focused on characterising the events at 7 dpi as we were interested in understanding the events taking place in the adipose tissue preceding tissue wasting. However, we have performed flow cytometry analyses at 7dpi (**Figure 6**) to examine specific cell populations in more detail.

8) Using scRNA-seq data the authors could consider exploring cell-cell interactions between transcriptomic profile of Tregs and Th17/Vgamma6+ cells, taking into account that Treg/ Th17 balance is involved in regulation of inflammation. The cell-cell interactions between Il17a expressing cells and preadipocytes could be investigated as well.

Authors: We thank the reviewer for this suggestion. We have previously explored interactions between *Il17a*-expressing cells and preadipocytes¹⁶, where we identified interactions that were predicted to promote alterations to lipid metabolism. Given that the balance between Tregs and TH17 is an important driver of tissue inflammation and pathology, we prioritised this interaction in this manuscript. We have now included a cell-cell interaction analysis between Tregs and TH17 cells, which is presented in supplementary Figure 7C in the revised version of the manuscript. In line with our hypothesis, cell-cell communication analyses predict an interaction between TH17-derived *Il17a* and Treg-derived *Il17ra*. We also observe strong predicted interactions between TH17-derived *Gas6* and Treg-derived *Axl* and *Mertk*. *Gas6-Axl* interactions have previously been shown to enhance the suppressive function of Tregs³². As such, we have included the following text in the Results sections to address this, as follow:

Line 391 “We next examined whether there was a cell-cell communication axis between TH17 cells and other replicative T cells found in the adipose tissue during infection (**Figure S7C**). NicheNet analysis predicted that TH17 cells produce several ligands that can be sensed by Tregs, including *Il17a* and *Gas6* (**Figure S7C**). Interestingly, *Gas6* interactions with *Axl* on Tregs has previously been shown to enhance the suppressive effects of Tregs, which could explain why *Il17ra*^{-/-} mice develop more severe clinical scores than wild type mice (**Figure 5B**).”

We have also added additional details to the Methods section to detail the use of the NicheNet package for cell-cell communication analyses:

Line 940 “Cell-cell interaction analysis mediated by ligand-receptor expression level was conducted using NicheNet³³ with default parameters using “mouse” as a reference organism, comparing differentially expressed genes between experimental conditions (condition_oi = “Infected”, condition_reference = “Uninfected”).”

References

1. Enerbäck, S. *et al.* Mice lacking mitochondrial uncoupling protein are cold-sensitive but not obese. *Nature* **387**, 90–94 (1997).
2. Ikeda, K. *et al.* UCP1-independent signaling involving SERCA2b-mediated calcium cycling regulates beige fat thermogenesis and systemic glucose homeostasis. *Nat Med* **23**, 1454–1465 (2017).
3. Gracey, E. *et al.* Sexual Dimorphism in the Th17 Signature of Ankylosing Spondylitis. *Arthritis & Rheumatology* **68**, 679–689 (2016).
4. Yu, J. J., Ruddy, M. J., Conti, H. R., Boonananantanasarn, K. & Gaffen, S. L. The Interleukin-17 Receptor Plays a Gender-Dependent Role in Host Protection against *Porphyromonas gingivalis*-Induced Periodontal Bone Loss. *Infect Immun* **76**, 4206–4213 (2008).
5. De Niz, M. *et al.* Organotypic endothelial adhesion molecules are key for *Trypanosoma brucei* tropism and virulence. *Cell Rep* **36**, 109741 (2021).
6. Kenna, T. J. & Brown, M. A. The role of IL-17-secreting mast cells in inflammatory joint disease. *Nat Rev Rheumatol* **9**, 375–379 (2013).
7. Barnie, P. A., Lin, X., Liu, Y., Xu, H. & Su, Z. IL-17 producing innate lymphoid cells 3 (ILC3) but not Th17 cells might be the potential danger factor for preeclampsia and other pregnancy associated diseases. *Int J Clin Exp Pathol* **8**, 11100–11107 (2015).
8. Ganeshan, K. *et al.* Energetic Trade-Offs and Hypometabolic States Promote Disease Tolerance. *Cell* **177**, 399–413.e12 (2019).
9. Jansson, P. A., Larsson, A., Smith, U. & Lönnroth, P. Glycerol production in subcutaneous adipose tissue in lean and obese humans. *J Clin Invest* **89**, 1610–1617 (1992).
10. Edwards, M. & Mohiuddin, S. S. Biochemistry, Lipolysis. in *StatPearls* (StatPearls Publishing, 2023).
11. Schweiger, M. *et al.* Measurement of Lipolysis. *Methods Enzymol* **538**, 171–193 (2014).
12. Kosteli, A. *et al.* Weight loss and lipolysis promote a dynamic immune response in murine adipose tissue. *J Clin Invest* **120**, 3466–3479 (2010).
13. Quintana, J. F. *et al.* $\gamma\delta$ T cells control murine skin inflammation and subcutaneous adipose wasting during chronic *Trypanosoma brucei* infection. *Nat Commun* **14**, 5279 (2023).
14. Xu, X., De Pergola, G. & Björntorp, P. The effects of androgens on the regulation of lipolysis in adipose precursor cells. *Endocrinology* **126**, 1229–1234 (1990).

15. Alter, D. & Koch, D. D. A Review of Acute Pancreatitis. *JAMA* **325**, 2402 (2021).
16. Quintana, J. F. *et al.* Spatially-resolved single cell transcriptomics reveal a critical role for $\gamma\delta$ T cells in the control of skin inflammation and subcutaneous adipose wasting during chronic *Trypanosoma brucei* infection. 2023.03.01.530674 Preprint at <https://doi.org/10.1101/2023.03.01.530674> (2023).
17. Zappia, L. & Oshlack, A. Clustering trees: a visualization for evaluating clusterings at multiple resolutions. *GigaScience* **7**, (2018).
18. Morgan, D. & Tergaonkar, V. Unraveling B cell trajectories at single cell resolution. *Trends in Immunology* **43**, 210–229 (2022).
19. Nguyen, H. T. T., Guevarra, R. B., Magez, S. & Radwanska, M. Single-cell transcriptome profiling and the use of AID deficient mice reveal that B cell activation combined with antibody class switch recombination and somatic hypermutation do not benefit the control of experimental trypanosomiasis. *PLoS Pathog* **17**, e1010026 (2021).
20. Zhang, C. *et al.* Single-cell sequencing reveals antitumor characteristics of intratumoral immune cells in old mice. *J Immunother Cancer* **9**, e002809 (2021).
21. Edwards, S. C. *et al.* PD-1 and TIM-3 differentially regulate subsets of mouse IL-17A–producing $\gamma\delta$ T cells. *Journal of Experimental Medicine* **220**, e20211431 (2022).
22. Balato, A., Unutmaz, D. & Gaspari, A. A. Natural Killer T Cells: An Unconventional T-Cell Subset with Diverse Effector and Regulatory Functions. *Journal of Investigative Dermatology* **129**, 1628–1642 (2009).
23. Quintana, J. F. *et al.* Single cell and spatial transcriptomic analyses reveal microglia-plasma cell crosstalk in the brain during *Trypanosoma brucei* infection. *Nat Commun* **13**, 5752 (2022).
24. Blasius, A. L. & Colonna, M. Sampling and signaling in plasmacytoid dendritic cells: the potential roles of Siglec-H. *Trends Immunol* **27**, 255–260 (2006).
25. Valente, M. *et al.* Novel mouse models based on intersectional genetics to identify and characterize plasmacytoid dendritic cells. *Nat Immunol* **24**, 714–728 (2023).
26. Li, J. *et al.* Mature dendritic cells enriched in immunoregulatory molecules (mregDCs): A novel population in the tumour microenvironment and immunotherapy target. *Clin Transl Med* **13**, e1199 (2023).
27. Bosteels, C. *et al.* Inflammatory Type 2 cDCs Acquire Features of cDC1s and Macrophages to Orchestrate Immunity to Respiratory Virus Infection. *Immunity* **52**, 1039-1056.e9 (2020).
28. Delobel, P., Ginter, B., Rubio, E., Balabanian, K. & Lazennec, G. CXCR2 intrinsically drives the maturation and function of neutrophils in mice. *Front Immunol* **13**, 1005551 (2022).
29. Zhang, Z. *et al.* Prmt1 upregulated by Hdc deficiency aggravates acute myocardial infarction via NETosis. *Acta Pharmaceutica Sinica B* **12**, 1840–1855 (2022).
30. Sprenkeler, E. G. G. *et al.* S100A8/A9 Is a Marker for the Release of Neutrophil Extracellular Traps and Induces Neutrophil Activation. *Cells* **11**, 236 (2022).
31. Bradley, L. M., Douglass, M. F., Chatterjee, D., Akira, S. & Baaten, B. J. G. Matrix metalloprotease 9 mediates neutrophil migration into the airways in response to influenza virus-induced toll-like receptor signaling. *PLoS Pathog* **8**, e1002641 (2012).
32. Zhao, G. *et al.* Growth Arrest-Specific 6 Enhances the Suppressive Function of CD4+CD25+ Regulatory T Cells Mainly through Axl Receptor. *Mediators Inflamm* **2017**, 6848430 (2017).
33. Browaeys, R., Saelens, W. & Saeys, Y. NicheNet: modeling intercellular communication by linking ligands to target genes. *Nat Methods* **17**, 159–162 (2020).

REVIEWERS' COMMENTS

Reviewer #1 (Remarks to the Author):

All my comments have been satisfactorily addressed.

Reviewer #3 (Remarks to the Author):

The authors answered all my questions and further significantly improved the manuscript. The latter provides very important piece of information adding to the understanding of trypanosomiasis. I have just one small remark and one curiosity question:

1) To distinguish DC1 from DC2 it would be nice to add a canonical Cd8a gene marker and for neutrophils Ly6g and Cd177, supplementary figure S6.

2) The KEGG data Fig 3C demonstrates absence of Th17 pathway in females despite the presence of IL17A in CyTOF and no significant differences between infected females and males Fig. 4J. In recently published scRNAseq study (Nature Communications 29 August 2023) authors show that there is no Th17 cells identified within the subcutaneous adipose tissue of female mice. Could the authors elaborate on these differences?

Decision on Nature Communications manuscript NCOMMS-23-23052A

We thank all of the reviewers for their second round of comments. We hope that our additional revisions satisfy the remaining concerns. Below, we have included point-by-point answers to the comments raised by reviewers 1 and 3 (in blue).

Reviewer #1 (Remarks to Authors)

All my comments have been satisfactorily addressed.

Authors: We thank this reviewer for taking their time to review our manuscript.

Reviewer #3 (Remarks to Authors)

The authors answered all my questions and further significantly improved the manuscript. The latter provides very important piece of information adding to the understanding of trypanosomiasis. I have just one small remark and one curiosity question:

- 1) To distinguish DC1 from DC2 it would be nice to add a canonical Cd8a gene marker and for neutrophils Ly6g and Cd177, supplementary figure S6.

Authors: We thank the reviewer for their helpful suggestion and we have now updated this figure to include Cd8a (cDC1) and Ly6g and Cd177 (neutrophils) in the dot plot, and have updated the manuscript as follows:

Line 368-373: “We classified these as *Ly6i⁺* macrophages (*Cd68⁺*, *Ms4a6d⁺*, *Ly6i⁺*, *Pla2g7⁺*)³⁶, plasmacytoid DCs (pDCs; *Siglech⁺*, *Cox6a2⁺*, *Ccr9⁺*, *Cd300c⁺*)^{39,40}, mature DCs (*Fscn1⁺*, *Cacnb3⁺*, *Socs2⁺*, *Tbc1d4⁺*)⁴¹, conventional DC2 (cDC2; *S100a4⁺*, *Gpr183⁺*, *H2-Eb1⁺*, *Marveld1⁺*)⁴², *Mrc1⁺* macrophages (*Mrc1⁺*, *Folr2⁺*, *Pf4⁺*, *Fcrls⁺*)³⁶, neutrophils (*Ly6g⁺*, *Cd177⁺*, *Cxcr2⁺*, *Hdc⁺*, *S100a9⁺*, *Mmp9⁺*)^{43–46}, and cDC1 (*Cd8a⁺*, *Xcr1⁺*, *Gcsam⁺*, *Clec9a⁺*, *Tlr3⁺*)⁴² (Supplementary Figure 6B).”

- 2) The KEGG data Fig 3C demonstrates absence of Th17 pathway in females despite the presence of IL17A in CyTOF and no significant differences between infected females and males Fig. 4J. In recently published scRNAseq study (Nature Communications 29 August 2023) authors show that there is no Th17 cells identified within the subcutaneous adipose tissue of female mice. Could the authors elaborate on these differences?

Authors: We thank the reviewer for their question regarding the commonalities between our current work and previous work. In our previous work (Nature Communications, 29th August 2023) we performed single cell sequencing on murine skin of female BALB/c mice. In this

study, we did, indeed, observe an expansion of *Cd4*⁺ T cells, but we did not detect *Il17a* expression, or cells that we could label as T_H17 cells within this compartment, which may be technical. In our study of the murine skin, we included all of the layers of the skin, including the subcutaneous white adipose tissue, but did not study the inguinal white adipose tissue, but we did not detect *Il17a* transcripts, which may be due to an excess of transcripts from other cell types, such as keratinocytes, and reaching the sequencing saturation limit. However, we were able to detect increased IL-17A expression at the protein level in gamma delta T cells in the murine skin during *T. brucei* infection. In the current study, we focused only on the inguinal white adipose tissue and had very few stromal cells, which may have enable deeper sequencing of the T cell compartment and detection of *Il17a* transcripts. We have also updated the manuscript with the following text, to acknowledge the similarities and differences with our previous study:

Lines 512-515: “In this study, we set out to determine the impact that African trypanosomes on the subcutaneous adipose tissue, which is adjacent to the skin and might be critical for forward transmission. In our recent work, we identified that skin *Il17a*⁺ V γ 6⁺ cells play an important role in controlling subcutaneous white adipose tissue wasting⁵⁴, and here we wanted to explore the role of IL-17A in greater detail, focusing exclusively on the iWAT.”